# The gut microbiome and early-life growth in a population with high prevalence of stunting

Ruairi C. Robertson[1,11,12], Thaddeus J. Edens[2,12], Lynnea Carr [3,12], Kuda Mutasa[4], Ethan K. Gough [5], Ceri Evans [1,4], Hyun Min Geum[6], Iman Baharmand [6], Sandeep K. Gill[6], Robert Ntozini [4], Laura E. Smith[4,7], Bernard Chasekwa[4], Florence D. Majo[4], Naume V. Tavengwa[4], Batsirai Mutasa[4], Freddy Francis [8], Joice Tome[4], Rebecca J. Stoltzfus[9], Jean H. Humphrey[5], Andrew J. Prendergast [1,4,5] & Amee R. Manges [6,10] ✉

Stunting affects one-in-five children globally and is associated with greater infectious morbidity, mortality and neurodevelopmental deficits. Recent evidence suggests that the early-life gut microbiome affects child growth through immune, metabolic and endocrine pathways. Using whole metagenomic sequencing, we map the assembly of the gut microbiome in 335 children from rural Zimbabwe from 1–18 months of age who were enrolled in the Sanitation, Hygiene, Infant Nutrition Efficacy Trial (SHINE; NCT01824940), a randomized trial of improved water, sanitation and hygiene (WASH) and infant and young child feeding (IYCF). Here, we show that the early-life gut microbiome undergoes programmed assembly that is unresponsive to the randomized interventions intended to improve linear growth. However, maternal HIV infection is associated with over-diversification and over-maturity of the early-life gut microbiome in their uninfected children, in addition to reduced abundance of *Bifidobacterium* species. Using machine learning models (XGBoost), we show that taxonomic microbiome features are poorly predictive of child growth, however functional metagenomic features, particularly B-vitamin and nucleotide biosynthesis pathways, moderately predict both attained linear and ponderal growth and growth velocity. New approaches targeting the gut microbiome in early childhood may complement efforts to combat child undernutrition.

Stunting, or linear growth failure, arises from a network of underlying factors including inadequate dietary quantity and quality, and affects 22% of children under 5 years of age worldwide[1,2]. Stunting is associated with infectious morbidity, reduced childhood survival and impaired cognitive development[3]. The lifelong impacts of poor growth contribute to an intergenerational cycle of stunting and impaired development, lower educational attainment, and reduced adult economic productivity[4]. Nutritional interventions, however, only reduce stunting by ~12%[5], suggesting that other pathophysiological mechanisms contribute to chronic undernutrition, which may inform new therapeutic strategies.

The determinants of stunting and other forms of child undernutrition are complex and include a myriad of biological, environmental and social factors including breastfeeding and complementary

feeding practices, household water, sanitation and hygiene (WASH) practices, birthweight, maternal HIV status, maternal anthropometry and maternal education. Growing evidence suggests that a subclinical disorder of the small intestine, termed environmental enteric dysfunction (EED), may also play a role in impaired child growth[6]. EED is characterized by blunted intestinal villi, increased gut permeability, and microbial translocation into the circulatory system resulting in both local and chronic inflammation and nutrient malabsorption[7,8]. It is hypothesized that high enteric pathogen carriage, as seen in poor-hygiene, low-resource settings, contributes to the pathophysiology of EED[9–11]; however, interventions to improve WASH and reduce the pathogen burden in children have failed to demonstrate improvements in linear growth[12]. Additionally, both enteric pathogen load and common biomarkers of EED are not consistently associated with linear growth in different geographical cohorts[13–16], suggesting that the pathway linking microbial exposures, impaired gut function and early-life growth remains to be fully elucidated.

In addition to research investigating the influence of diarrhoeal pathogens on child undernutrition and EED, emerging evidence supports the role of the commensal gut microorganisms in mediating child growth. Healthy-growing children exhibit a patterned ecological assembly of the gut microbiome through the first 2 years of life, which is defined by delivery mode, breastfeeding, complementary feeding practices and geography[17–20]. This microbial succession impacts a number of metabolic, immune and endocrine pathways in early life that contribute to early-life growth and development[21]. Disturbances to this normal microbiome maturation therefore may impair these critical growth and developmental pathways. Immaturity of the early-life gut microbiome is associated with severe acute malnutrition[22], whilst reduced microbiome diversity is associated with higher risk of future diarrheal episodes[23]. Indeed, a 'malnourished' early-life gut microbiome can recapitulate phenotypes of faltering growth and EED when transplanted into germ-free mice and pigs[24,25]. Furthermore, nutritional interventions designed to specifically target the impaired gut microbiome in acute malnutrition in both animal studies and small-scale human trials have recently demonstrated a positive effect on ponderal growth[26,27], but not on linear growth.

Microbiome differences that may contribute to stunting are likely influenced by a number of environmental factors including household WASH, infant feeding practices and maternal HIV infection. To date, little research has investigated the effect of improved WASH or infant feeding interventions on the assembly of the infant gut microbiome in low resources settings. However, recent data show that children who are HIV-exposed but uninfected (CHEU), consume breast-milk with an altered oligosaccharide composition from their mothers[28], and may be exposed to abnormal microbiome profiles from their mothers, which have been reported in people living with HIV[29,30]. CHEU also receive prophylactic antibiotics, to prevent infectious morbidity associated with HIV exposure. Each of these exposures may influence the seeding and succession of the gut microbiome in CHEU[31,32], which may contribute to the high prevalence of stunting observed in CHEU[33]. Evidence of the effect of other early-life environmental exposures on the assembly of the infant gut microbiome in low resources settings is scarce but may provide insights into the influence of microbial and microbiota-modifying exposures on child growth in the context of undernutrition.

Previous cross-sectional data from sub-Saharan Africa hypothesized that decompartmentalization of the gastrointestinal tract occurs in stunted children, as demonstrated by the overgrowth of oropharyngeal bacterial taxa in the intestine[34,35], whilst a handful of other cross-sectional studies report variations in gut microbiota composition in stunted children that are inconsistent across geographical settings[36–38]. We previously reported that the maternal gut microbiome can predict birthweight and neonatal growth in rural Zimbabwe[30]. However, there are few studies mapping the compositional and functional maturation of the gut microbiome throughout early childhood, accounting for feeding, WASH, maternal HIV infection and other environmental exposures, in populations from low-resource settings and at high risk of stunting.

Here, we characterize the succession and maturation of the fecal microbiome from 1–18 months of age in 335 children from rural Zimbabwe who were enrolled in the Sanitation, Hygiene, Infant Nutrition Efficacy (SHINE) Trial[39]. We hypothesized that randomized nutrition and hygiene interventions would alter gut microbiome development and that gut microbiome composition and function could predict child growth. We show that the early-life gut microbiome undergoes programmed assembly that is unresponsive to the randomized interventions intended to improve linear growth. Maternal HIV infection is associated with over-diversification and over-maturity of the early-life gut microbiome in their uninfected children in addition to reduced abundance of *Bifidobacterium* species. Using machine learning models (XGBoost), we show that taxonomic microbiome features are poorly predictive of growth; however, functional metagenomic features, particularly B-vitamin and nucleotide biosynthesis pathways, moderately predict both attained linear and ponderal growth and growth velocity through the first 18 months of life.

## Results

### Sub-study population characteristics

The fecal microbiota was characterized in 875 samples from 335 children from 1–18 months of age (Supplementary Fig. 1). A mean (SD) of 2.6 (1.3) samples were analysed per child (Supplementary Table 1). The children in the microbiome sub-study largely resembled the population of all live-born infants in the overall SHINE trial cohort (Supplementary Table 2); however, the microbiome sub-study included a larger number of children who were born to women living with HIV (29.6%) compared to the whole SHINE cohort (15.6%), due to the deliberate over-sampling of mothers living with HIV and their infants. In addition, the microbiome sub-study included infants with slightly older mothers and longer gestational ages. The majority of infants were born by vaginal delivery (94.5%) in an institution (89.9%) and were exclusively breastfed (91% at 3 months). Prevalence of stunting (length-for-age Z-score (LAZ) < −2) varied from 18–34% across study timepoints. Only 2–8% of HIV-negative mothers reported infant antibiotic use across visits, compared with 55–76% of HIV-positive mothers, which was largely attributed to infant cotrimoxazole use as part of WHO guidelines for HIV-exposed children.

### Metagenome sequencing performance

Overall, 875 unique whole metagenome sequencing datasets were used. On average, 12 million ± 4.2 million quality-filtered read pairs were generated per sample. Sixteen negative controls produced a mean of 655 quality-filtered reads (range = 149 to 1,425; SD = 456). The median percent of human reads detected was 0.05% but ranged widely by age group and decreased over time (Supplementary Fig. 2a). The median percent un-annotatable reads detected in each sample was 58.6% and increased over time (Supplementary Fig. 2b). Thirty-six samples were subject to repeated extraction and metagenome sequencing to assess technical variation. These samples originated from 4 unique children, each with 3 visit samples, where each visit sample was extracted and sequenced in 3 replicates. Principal coordinates analysis (PCoA) of Bray-Curtis distances and phylum-level relative abundances revealed little variation between replicates (Supplementary Fig. 2c, d).

### Succession of gut microbiome composition in early childhood

After prevalence and relative abundance threshold filtering, 161 annotated bacterial species were identified. Seven Eukaryotic and 4 Archaeal species were detected in a small proportion of samples, but these did not meet the prevalence thresholds (Supplementary

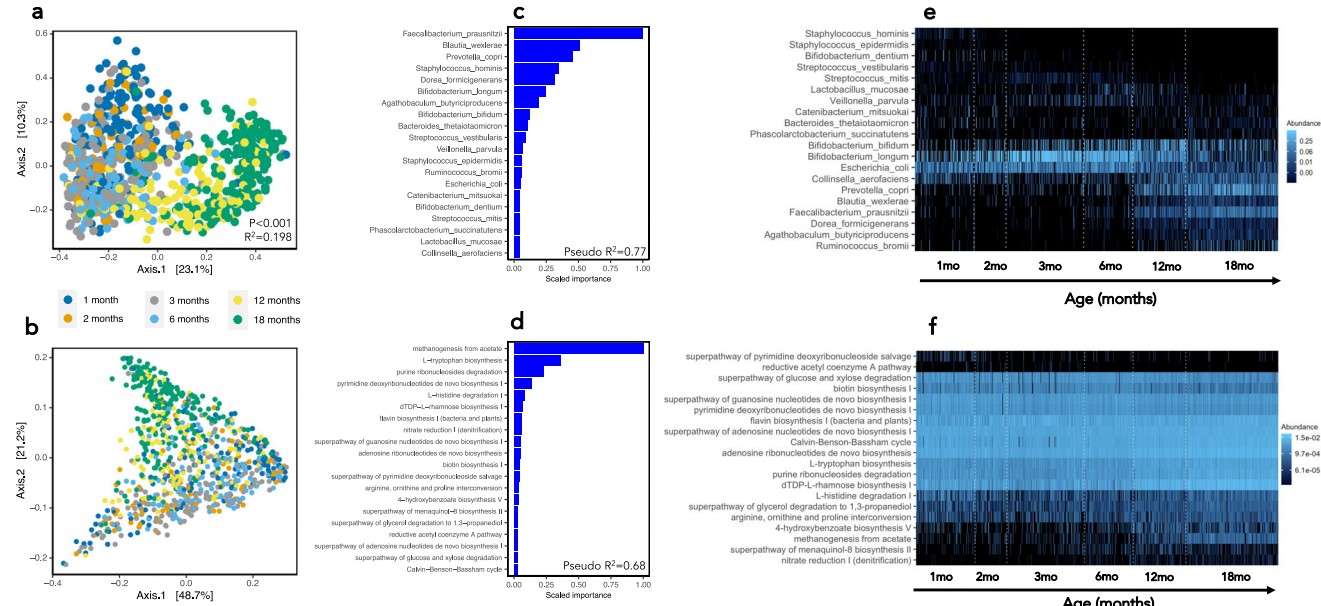

**Fig. 1 | Compositional and functional maturation of the gut microbiome of 335 infants from rural Zimbabwe from 1–18 months of age.** PCoA of Bray-Curtis distances of species (**a**; PERMANOVA; two-sided, $p < 0.001$) and metagenomic pathways (**b**; PERMANOVA; two-sided, $p < 0.001$) coloured by age category. The top

20 features and model pseudo-$R^2$ from XGBoost models predicting age using species (**c**) or pathways (**d**) are ranked by scaled feature importance and relative abundance(0–1) plotted by age (**e**, **f**) to visualize taxonomic and functional microbiome succession from 1–18 months of age. $n = 875$ samples.

Table 3). *Bifidobacterium longum* was the predominant species at all time-points up to 12 months of age. Four other *Bifidobacteria* speces (*B. breve, B. bifidum, B. pseudocatenulatum, and B. kashiwanohense*), *Escherichia coli, Bacteroides fragilis* and *Veillonella* species were consistently amongst the most abundant species at the earlier time points before being outnumbered by *Faecalibacterium prausnitzii* and *Prevotella copri* at 12 and 18 months of age. Taxonomic α-diversity metrics and gene richness tended to decline or remain stable over the first 4–6 months of life, during exclusive breastfeeding, but increased as expected with infant age from 6–18 months of age (Supplementary Fig. 3a, b), with the introduction of complementary feeds. A large proportion of variation in both compositional (PERMANOVA; $R^2 = 0.198$, $P < 0.001$) and functional β-diversity ($R^2 = 0.144$, $P < 0.001$) was explained by age category (Fig. 1a, b and Supplementary Fig. 3c, d).

Age is the most influential variable defining microbiome composition and function in early childhood[17]. We employed extreme gradient boosting machines (XGBoost), a machine learning approach, to train and test a model of compositional and functional microbiome maturation, with child age as an outcome. Children who were born to HIV-negative mothers, who were non-stunted at 18 months (LAZ > −2) and had at least 2 stool samples collected were used as a 'healthy training set' ($n = 265$ samples and 97 infants), which was then used to predict child age in a 'healthy test set' ($n = 66$ samples and 66 infants) and an 'unhealthy test set' ($n = 528$ samples and 169 infants). Using species composition, the microbiome was highly predictive of child age (Model pseudo-$R^2 = 0.77$, Mean Absolute Error [MAE] = 1.4 months). This 'microbiota age' score was also strongly correlated with chronological age in the independent subset of children from the 'healthy' test set who were also non-stunted and born to HIV-negative mothers (pseudo-$R^2 = 0.65$). The species most strongly predictive of age included *Faecalibacterium prausnitzii, Blautia wexlerae, Prevotella copri, Staphylococcus hominis, Dorea formicigenerans, Bifidobacterium longum, Agathobaculum butyriciproducens, Bifidobacterium bifidum, Bacteroides thetaiotaomicron, Streptococcus vestibularis* and *Veillonella parvula* (Fig. 1c). Metagenomic pathways also predicted age with high accuracy (Model pseudo-$R^2 = 0.68$; MAE = 1.5; Fig. 1d). The pathways most strongly predictive of age included methanogenesis from acetate

(METH-ACETATE-PWY), multiple nucleotide and amino acid metabolic pathways, including L-tryptophan biosynthesis (TRPSYN-PWY), purine ribonucleosides degradation (PWY0-1296), pyrimidine deoxyribonucleotides de novo biosynthesis I (PWY-7184), L-histidine degradation I (HISDEG-PWY), dTDP-L-rhamnose biosynthesis I (DTDPRHAMSYN-PWY), flavin biosynthesis I (RIBOSYN2-PWY) and nitrate reduction I (DENITRIFICATION-PWY). This 'metagenome age' was also correlated with age in the independent subset of children from the 'healthy' test set who were also non-stunted and born to HIV-negative mothers (pseudo-$R^2 = 0.56$). Using these models, we created a microbiota-for-age Z-score (MAZ) and metagenome-for-age Z-score (MetAZ), which accounted for variance of microbiota ages with respect to chronological ages at each study visit (see Methods). The top 20 features contributing most strongly to age predictions are plotted in Fig. 1e, f. In summary, both composition and function of the gut microbiome are highly predictive of child age from 1–18 months, suggesting a patterned, assembly of the gut microbiome in this setting.

## WASH and IYCF interventions have little impact on the infant gut microbiome

We have previously reported in the SHINE trial that the WASH intervention had no impact on infant growth, whilst the IYCF intervention increased LAZ scores by 0.16 in HIV-unexposed children and 0.26 in CHEU, leading to a 20–23% relative reduction in stunting by 18 months of age. We tested whether these randomized interventions impacted any metrics of gut microbiome diversity or maturity in each age group. By performing PCoA on Bray-Curtis distances of taxonomic data and functional data, we found no significant differences in β-diversity between IYCF and non-IYCF arms at any time-point. There were some weak differences in Bray-Curtis distances for microbiome composition between WASH and non-WASH arms at the 3-month time-point (PERMANOVA, $P = 0.041$; $R^2 = 0.009$) and 18-month time-points ($P = 0.02$; $R^2 = 0.01$), but at no other time-points (Fig. 2a–d). No significant differences were observed in α-diversity metrics or gene richness between WASH versus non-WASH arms nor IYCF versus non-IYCF arms at any time-point (Fig. 2e, f). Using multivariable regression analyses (MaAsLin2), and following adjustment for covariates (age in days at

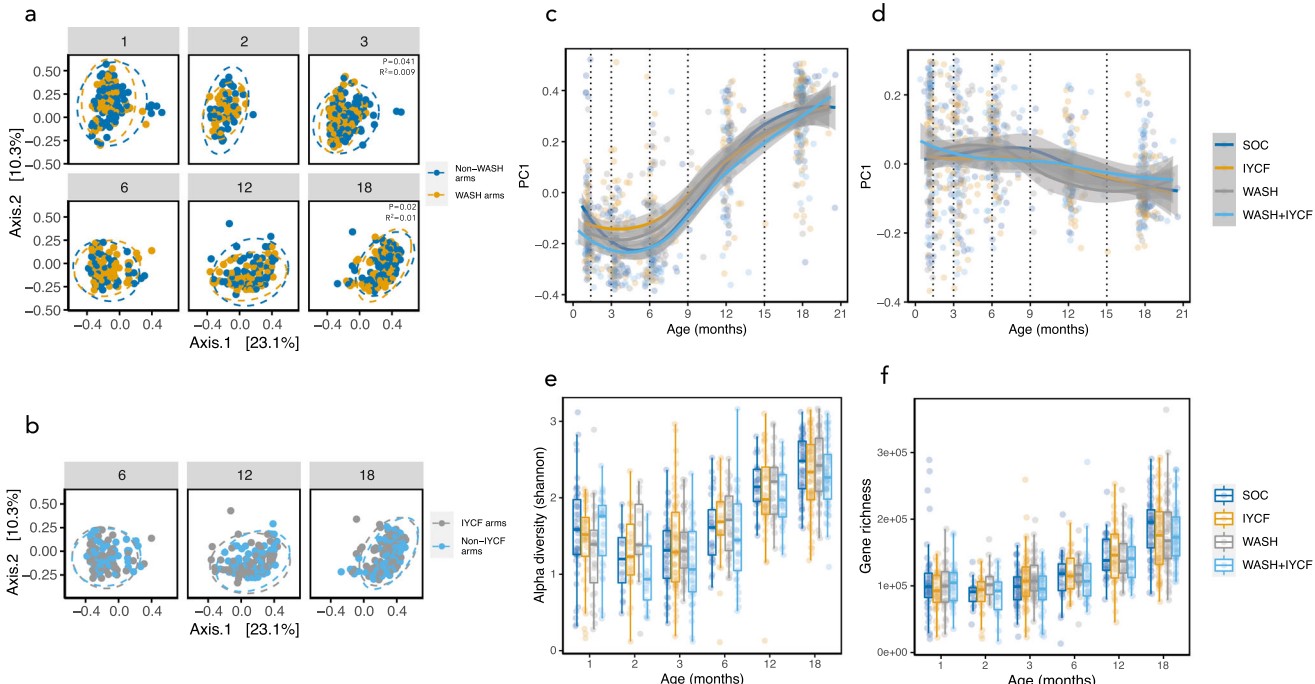

**Fig. 2 | Impact of randomized WASH and IYCF interventions on infant gut microbiome.** PCoA of Bray-Curtis distances species composition coloured by WASH vs non-WASH arms (**a**), including PERMANOVA model results, and IYCF vs non-IYCF arms (**b**) are plotted (dotted lines represents 95% confidence ellipses; two-sided p-value) in addition to the first component (PC1) from PCoA of species (**c**) and pathways (**d**; lines represent smoothed conditional means and grey shaded areas represent 95% confidence intervals). The IYCF intervention was introduced after 6 months of age, therefore direct comparisons of IYCF vs non-IYCF arms are not shown in the 1-, 2- and 3-month age categories. No significant differences were observed in Shannon alpha diversity (**e**) and gene richness (**f**) according to trial arm (the band indicates the median, the box indicates the first and third quartiles and the whiskers indicate ±1.5 × interquartile range). $n = 875$ samples. WASH water sanitation and hygiene arm, IYCF infant and young child feeding arm, WASH + IYCF combined WASH and IYCF arm, SOC standard of care arm, Non-WASH the two arms that did not contain WASH interventions, Non-IYCF the two arms that did not contain IYCF interventions.

stool sample collection, exclusive breastfeeding status, delivery mode, and randomised trial arm) there were also no differences in the relative abundance of species or pathways between intervention arms apart from a small number of features at 3 months in the WASH arms (increased *Klebsiella pneumoniae*, reduced *Collinsella aerofaciens* and more abundant metagenomic pathways involved in biotin and folate synthesis) and at 18 months in the IYCF arms (reduced *Eubacterium siraeum*, *E. rectale* and *Agathobaculum butyriciproducens*; Supplementary Data File 1 and Supplementary Data File 2). The SHINE WASH and IYCF had little impact on infant microbiome composition or function.

## HIV exposure is strongly associated with infant gut microbiome composition and function

We previously reported in this cohort that maternal HIV exposure significantly impacts infant growth, whereby CHEU displayed significantly poorer linear growth compared with children who are HIV-unexposed (CHU)[30]. We assessed diversity metrics and microbiome maturity in CHEU versus CHU adjusting for covariates ($n = 847$ stool samples total; children with 'unknown' or 'positive' HIV status at 18 months were excluded). The proportion of CHEU receiving prophylactic cotrimoxazole (caregiver-reported) ranged from 56–83% across the 3, 6, 12 and 18-month study visits, and the majority of CHEU were exclusively breastfed to age 6 months. Using multivariate regression analyses adjusting for exclusive breast-feeding status, delivery mode, age in days at stool sample collection and trial arm, we found that CHEU displayed significantly greater alpha diversity (Shannon index $\beta = 0.28$, $P = 0.01$; Evenness $\beta = 0.06$, $P = 0.02$) compared with CHU at 12 months of age (Fig. 3a, b); we used a permutation test, as the sample size was imbalanced in this age category ($n = 91$ CHEU, $n = 27$ CHU) and the significant differences persisted (Shannon

$P = 0.002$; Evenness $P = 0.022$). Metrics of α-diversity were significantly lower in CHEU at 2 months of age (evenness $\beta = -0.09$, $P = 0.038$). Metagenomic gene richness was also elevated in CHEU at 1, 6 and 12 months after adjusting for covariates ($P = 0.002$, $P = 0.006$, $P = 0.025$ respectively; Fig. 3c). Analysis of taxonomic $\beta$-diversity between CHEU and CHU also revealed significant differences at 1, 3, 6, 12 and 18 months suggesting that *in utero* HIV exposure was significantly associated with differences in gut microbiome succession and development throughout the first 18 months of life (Fig. 3d, e). HIV exposure also explained a significant proportion of the variation in metagenome pathway $\beta$-diversity (PERMANOVA; $R^2 = 0.035$; $P = 0.008$, two-sided) at 1 month of age (Fig. 3h, i), but not at later ages. We next tested the association between HIV exposure and gut microbiome maturity. We found that CHEU displayed greater microbiome age and MAZ, and hence microbiota over-maturity, compared with CHU at 6 months (6.0 vs 5.4 months; multivariate regression; $\beta = 1.58$, $P = 0.037$, two-sided) and 12 months of age (median 14.7 vs 9.2 months; $\beta = 3.05$, $P = 0.005$; Fig. 3f, g). However, at 18 months CHEU displayed lower microbiome age (15.3 vs 16.4 months; $\beta = -1.2$, $P = 0.023$). Similarly, CHEU displayed significantly greater metagenome ages and MetAZ scores compared with CHU at 1 month ($\beta = 0.89$, $P = 0.05$) and 6 months of age ($\beta = 1.7$, $P = 0.039$; Fig. 3j, k) after adjustment for covariates, suggesting that HIV exposure drives both compositional and functional microbiome over-maturity. Taken together, children born to mothers living with HIV and exposed to HIV during pregnancy exhibited greater compositional and functional microbiome diversity and over-maturity, especially in the first 12 months.

We next explored which species were differentially abundant between CHEU and CHU by performing multivariable regression analyses, adjusting for important confounding factors (age in days at stool sample collection, exclusive breastfeeding status, delivery mode, and

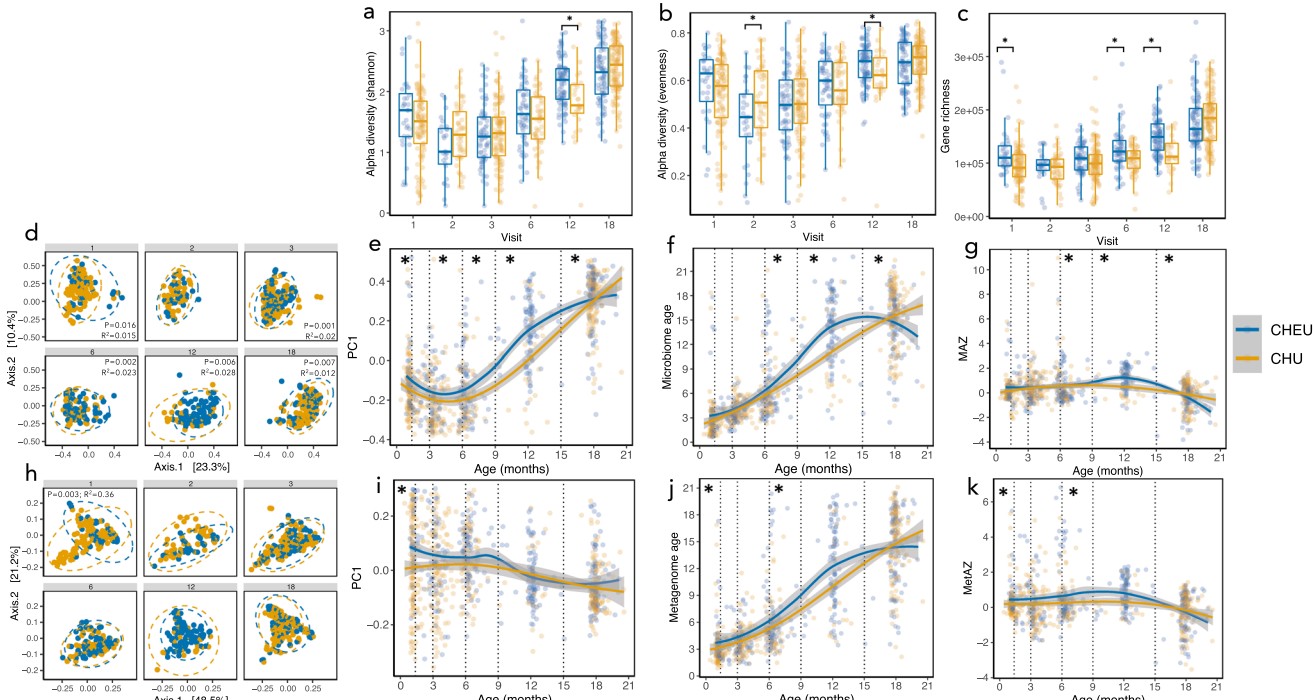

**Fig. 3 | Maternal HIV infection comprehensively alters infant gut microbiome diversity and maturity.** Shannon alpha diversity (**a**), species richness (**b**) and gene richness (**c**) shows significant over-diversification in CHEU vs CHU (multivariate regression, two-sided *p* < 0.05; the band indicates the median, the box indicates the first and third quartiles and the whiskers indicate ±1.5 × interquartile range). PCoA of Bray-Curtis distances (**d**) and PC1 (**e**) of species composition in CHEU vs CHU show significant differences throughout 18 months of life (PERMANOVA; two-sided *p*-value; dotted lines represent 95% confidence ellipses). Microbiome age (**f**) and microbiome-for-age Z score (MAZ; **g**) shows significant differences in gut microbiome maturity in CHEU vs CHU (multivariate linear regression analyses; *p* < 0.05, two-sided). PCoA of microbiome gene pathways shows differences in CHEU vs CHU at 1 month of age (**h**, **i**; PERMANOVA, two-sided *p*-value) in addition to differences in metagenome-for-age Z-score (MetAZ) at 1 and 6 months of age (**j**, **k**; multivariate linear regression analyses; two-sided, *p* < 0.05). Lines represent smoothed conditional means and grey shaded areas represent 95% confidence intervals. *n* = 859 samples. *p* < 0.05. CHEU children who are HIV-exposed but uninfected, CHU children who are HIV-unexposed and uninfected, PC1 principle component 1.

randomised trial arm). Between 1–3 months of age, two Bifidobacteria species, *B. longum* and *B. bifidum* (Fig. 4a, b), in addition to *Veillonella seminalis* were significantly less abundant in CHEU versus CHU (multivariate regression; *q* < 0.1; Supplementary Data File 1). Conversely, *Flavonifractor plautii* was significantly more abundant in CHEU at 3 months (*q* = 0.02). At 18 months, *B. bifidum* was again significantly less abundant in CHEU (*q* = 0.04), whilst two other Bifidobacteria species were also weakly associated with CHEU, whereby *B. breve* was lower and *B. pseudocatenulatum* higher (both *q* = 0.25). Regression analyses of metagenomic pathways with CHEU status generated similar outcomes. Following adjustment for covariates, a number of significant, yet weak, associations between HIV exposure status and metagenomic pathways were present at 1 and 3 months of age. At 1 month of age, these included significant negative associations between CHEU and amino acid synthetic pathways (superpathway of L-threonine biosynthesis, superpathway of L-isoleucine biosynthesis I and L-lysine biosynthesis I; Fig. 4c–e) and positive associations with pathways involved in the degradation of sugar derivatives, including fructuronate, glucoronate and galacturonate (PWY 7242 D-fructuronate degradation, PWY 6507 4-deoxy-L-threo-hex-4-eno-pyranuronate degradation, GALACTUROCAT PWY D-galacturonate degradation I, GALACT GLUCUROCAT PWY superpathway of hexuronide and hexuronate degradation and GLUCUROCAT PWY superpathway of beta D-glucuronide and D-glucuronate degradation; Fig. 4f–h, Supplementary Data File 2). A handful of pathways involved in fatty acid oxidation and fermentation (fatty acid beta oxidation peroxisome and succinate fermentation to butanoate) were also significantly positively associated with CHEU at 3 months of age. Collectively, these data suggest that maternal HIV infection may disrupt the

early-life gut microbiome, leading to reduced Bifidobacteria, reduced amino acid biosynthesis and elevated sugar degradation.

## Microbiome functionality, but not composition, predicts linear and ponderal growth

We next examined the relationship between taxonomic and functional features of the gut microbiome and attained growth (length-for-age Z-score [LAZ] and weight-for-height Z-score [WHZ]) and growth velocity (WHZ and LAZ velocity, increase in z-score increments per day between visits) from 1–18 months of age using XGBoost models. These models were used to test whether the composition or function of the gut microbiome could predict child growth. Models were run separately for each of the six age groups, stratified by maternal HIV status and run in two combinations of predictive features: (i) microbiome features alone (species or pathways); and (ii) microbiome features and epidemiological variables, which included maternal anthropometry, baseline WASH and infant diet variables, amongst others (Supplementary Data File 3). In models combining microbiome features with epidemiological features, birthweight, maternal height, maternal mid-upper arm circumference, and household wealth were all important predictors of attained infant LAZ and WHZ and growth velocity. Models including microbiome taxonomic features (species) alone performed poorly for both attained and growth velocity at every age category and regardless of HIV exposure status (Fig. 5a), with a majority of models resulting in pseudo-*R²* values <0. Model performance for linear growth improved when epidemiological features were included, suggesting that gut microbiota composition alone was poorly predictive of linear growth. Taxonomic features were weakly predictive of WHZ velocity at 2 and 3 months (pseudo-*R²* 0.19 and

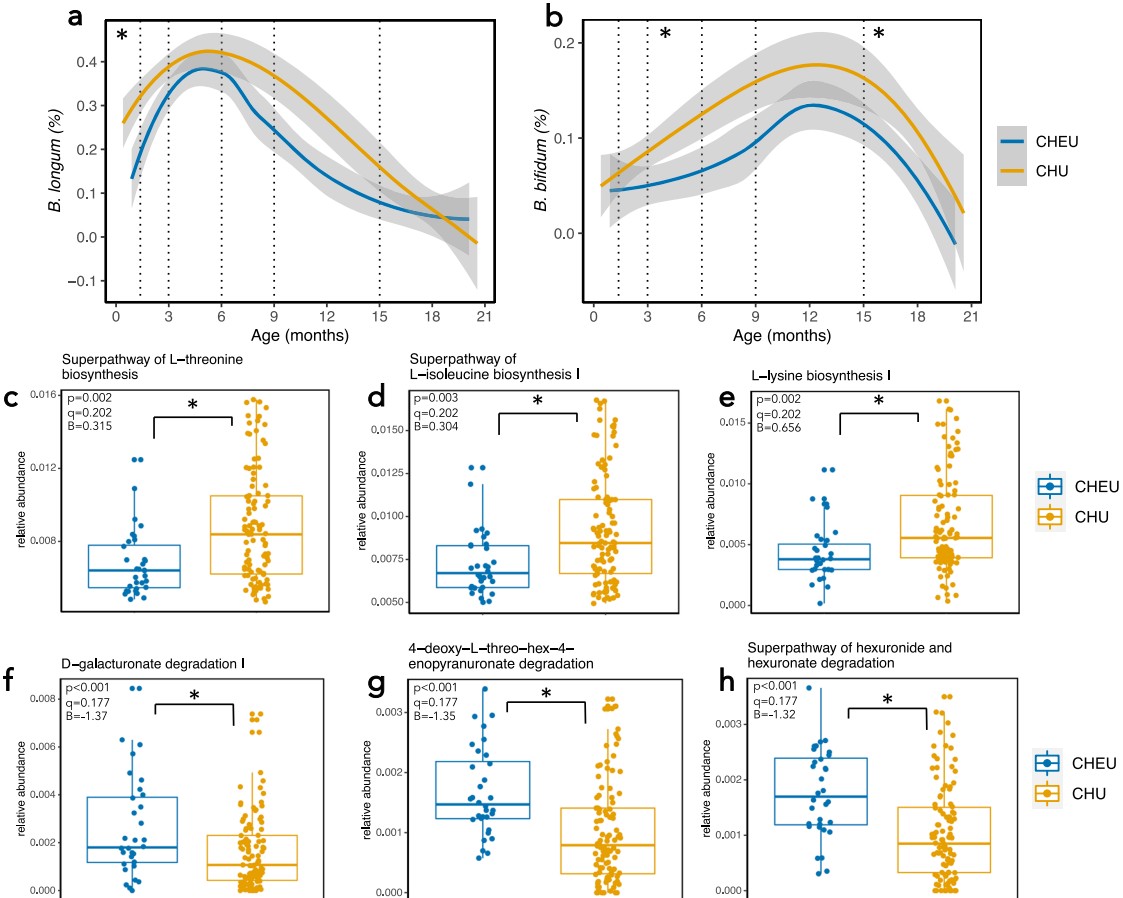

**Fig. 4 | Maternal HIV infection is associated with reduced abundance of Bifidobacteria abundance and amino acid biosynthesis genes.** Relative abundance(0–1) of *Bifidobacterium longum* (**a**) and *B. bifidum* (**b**) in the gut microbiome of CHEU and CHU at each age category via multivariate regression analyses (*$q < 0.1$; $n = 859$ samples total. Lines represent smoothed conditional means and grey shaded areas represent 95% confidence intervals). Multivariate regression of gene pathways demonstrates reduced abundance of amino acid biosynthetic pathways (**c**–**e**) and increase in abundance of pathways involved in degradation of sugar derivatives at 1 month of age (**f**–**h**; $n = 32$ CHEU, $n = 107$ CHU; multivariate linear regression analyses adjusted for multiple comparisons using Benjamini–Hochberg correction; the band indicates the median, the box indicates the first and third quartiles and the whiskers indicate ±1.5 × interquartile range; two-sided p-value). Significance thresholds defined using MaAsLin2 defaults (*$p < 0.05$, $q < 0.25$).

0.08) and LAZ velocity at 12 months (pseudo-$R^2 = 0.25$), but only in children born to HIV-negative mothers (Figs. 5a and 6a). Conversely, models containing functional metagenomic pathways were moderately predictive of both attained and future growth from 1–18 months of age (pseudo-$R^2 = 0$–0.66; Figs. 5a and 6a; $n = 856$ samples and $n = 854$ for LAZ and WHZ models respectively) albeit with relatively large MAE scores for both linear (0.54–0.99 LAZ) and ponderal growth models (0.71–1.15 WHZ). The inclusion of epidemiological variables in the metagenomic models added little to performance suggesting that pathway features were independently predictive of both linear and ponderal growth. MAE decreased in all models as age increased (Supplementary Fig. 4a–d). Models predicted WHZ better than LAZ and models including children born to HIV-negative mothers also tended to perform better. Hence, microbiome functional pathways moderately predicted child growth.

### Microbiome features associated with linear growth

Birthweight contributed most strongly to prediction of attained LAZ in a majority of models. Metagenomic pathways consistently performed better as predictors than other epidemiological features including maternal height. The most predictive pathways were largely similar between infants born to HIV-positive and HIV-negative mothers. At 1 month and 2-months, metagenomic pathways encoding purine and pyrimidine biosynthesis, lipid biosynthesis and biosynthesis of B

vitamins were consistently predictive of both attained LAZ and LAZ velocity (Fig. 5b). Accumulated local effects (ALE)[40,41] plots show the average effect of some of the most important features on model outcomes (Fig. 5c). At 3 months and 6 months, pathways encoding fermentation and carbohydrate biosynthesis were consistently predictive of attained LAZ and LAZ velocity, whilst carbohydrate and amino acid degradation pathways, amongst others, were predictive of growth at the oldest age groups. In particular, pathways involved in vitamin B biosynthesis (flavin, folate, biotin, thiazole and cobalamin biosynthetic pathways) were consistently predictive of attained LAZ, and included flavin biosynthesis I, 6-hydroxymethyl-dihydropterin diphosphate biosynthesis, superpathway of thiamin diphosphate biosynthesis, adenosylcobalamin salvage from cobinamide I, and thiazole biosynthesis I. Increasing abundances of B vitamin biosynthesizing genes contributed to increasing predicted growth in ALE plots, apart from thiamin biosynthesis, which predicted lower LAZ at 12 months (Fig. 5c). At 12 months of age, 4-coumarate degradation (anaerobic), a pathway involved in plant polysaccharide degradation, was the most predictive pathway of LAZ in children born to HIV-positive mothers, with greater abundance associated with greater LAZ. We also assessed pathways predicting growth velocity and found similar results to that of attained growth (Supplementary Fig. 5a). At 2 months of age, folate biosynthetic pathways (folate transformations II and N10-formyl-tetrahydrofolate biosynthesis) were amongst the top predictive

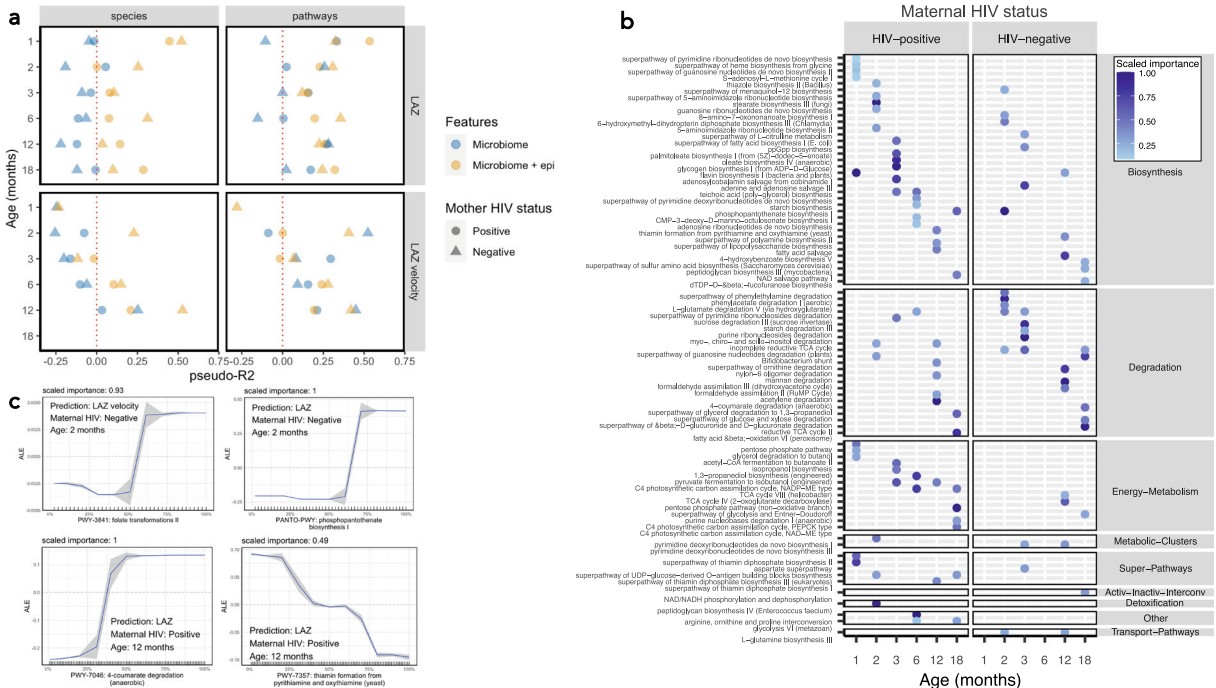

**Fig. 5 | Prediction of attained LAZ and LAZ velocity using XGBoost models.** Performance of XGBoost models as assessed by pseudo-$R^2$ values for prediction of attained LAZ and LAZ velocity (LAZ increase per day to next study visit) using species or metagenomic pathways, stratified by age category and maternal HIV status (**a**). Models were run using microbiome features alone (species or metagenomic pathways; blue points) and in combination with epidemiological variables (yellow points). The top ranked pathways predicting LAZ at each age category are plotted (**b**), stratified by maternal HIV status and coloured by scaled importance in the XGBoost model (darkness of blue shading indicates feature ranking in the model). Accumulated effect plots (ALE) of representative pathways ranking highly

in XGBoost model predictions display change in predicted linear growth (LAZ or LAZ velocity) by percentile of the feature abundance distribution (**c**). Tick marks on the x-axis are a rug plot of individual feature abundance percentiles. ALEs were generated using the *ALEplot* package and were plotted using *ggplot2*. Standard deviations (sd) were calculated per increment in microbiome feature and were used to calculate and plot increment-wise 95% confidence intervals as the average change in the outcome ±1.96(sd/sqrt(n)), where *n* is the number of observed feature values, and sd is the standard deviation of the change in the outcome variable in an interval. *n* = 856 and *n* = 460 samples for models predicting LAZ and LAZ velocity respectively.

features of linear growth velocity in children born to HIV-negative mothers, whilst at 3 months, purine and pyrimidine pathways were amongst the most predictive features of linear growth velocity. Amino acid and fatty acid biosynthetic pathways were strongly predictive of growth, as in the attained growth models, and glycogen biosynthesis pathways were consistently predictive of linear growth velocity at all ages from 2 months onwards. In summary, vitamin B biosynthesis, purine and pyrimidine and lipid biosynthetic pathways were all important pathways for infant linear growth prediction in early infancy, whilst carbohydrate degradation pathways were predictive of growth from 6 months onwards.

**Microbiome features associated with ponderal growth**

In the few models incorporating taxonomic features that weakly predicted WHZ and WHZ velocity, *Escherichia coli*, *Bacteroides fragilis* and *Megasphaera micronuciformis* were amongst the most predictive features between 1–3 months of age (Supplementary Fig. 6). Many of the same categories of biosynthetic microbiota pathways were predictive of both WHZ and LAZ, including amino acid and nucleotide (especially purine biosynthesis) biosynthetic pathways in addition to a number of lipid synthesis pathways at older age groups (Fig. 6b, c). The most predictive pathways were largely similar between infants born to HIV-positive and HIV-negative mothers. O-antigen biosynthesis pathways (PWY-7328 UDP-glucose-derived O-antigen building blocks biosynthesis & PWY-7332 UDP-N-acetylglucosamine-derived O-antigen building blocks biosynthesis and OANTIGEN-PWY pathway), which did not appear in LAZ models, were consistently amongst the most predictive features of attained WHZ and WHZ velocity at 1, 3, 6 and 12 months, whereby greater abundance was associated with reduced growth.

Pyrimidine and purine synthetic pathways were consistently the strongest predictors of WHZ velocity (albeit with varying directions of effect), including superpathway of pyrimidine deoxyribonucleotides de novo biosynthesis (PWY0-166) which was the strongest predictive feature of WHZ velocity at 12 months in children born to HIV-negative mothers (Supplementary Fig. 5b). Similar to the attained WHZ models, O-antigen biosynthesis pathways, amino acid synthetic pathways and glycogen biosynthesis pathways were all strongly predictive of ponderal growth velocity. Hence, metagenomic amino acid, nucleotide and O-antigen biosynthesis pathways are predictive of ponderal growth in this cohort.

## Discussion

We report the succession and maturation of the early-life gut microbiome in a cohort of 335 children from rural Zimbabwe from 1–18 months of age. We find that taxonomic composition of the gut microbiome is poorly predictive of child growth; however, functional composition moderately predicts both attained LAZ/WHZ and LAZ/WHZ velocity, with pathways including B vitamin and nucleotide biosynthesis genes amongst the most predictive of child growth. We also report that randomized WASH and IYCF interventions have little impact on early-life gut microbiome composition, whilst maternal HIV infection, which is associated with impaired infant growth, is associated with over-maturity of the gut microbiome, featuring a depletion in *Bifidobacteria* species and amino acid synthetic pathways. Collectively, these data suggest that disturbances in the functional potential of the infant gut microbiome may contribute to poor infant growth and that interventions targeting the infant gut microbiome may serve as novel solutions to combat child stunting, particularly in CHEU.

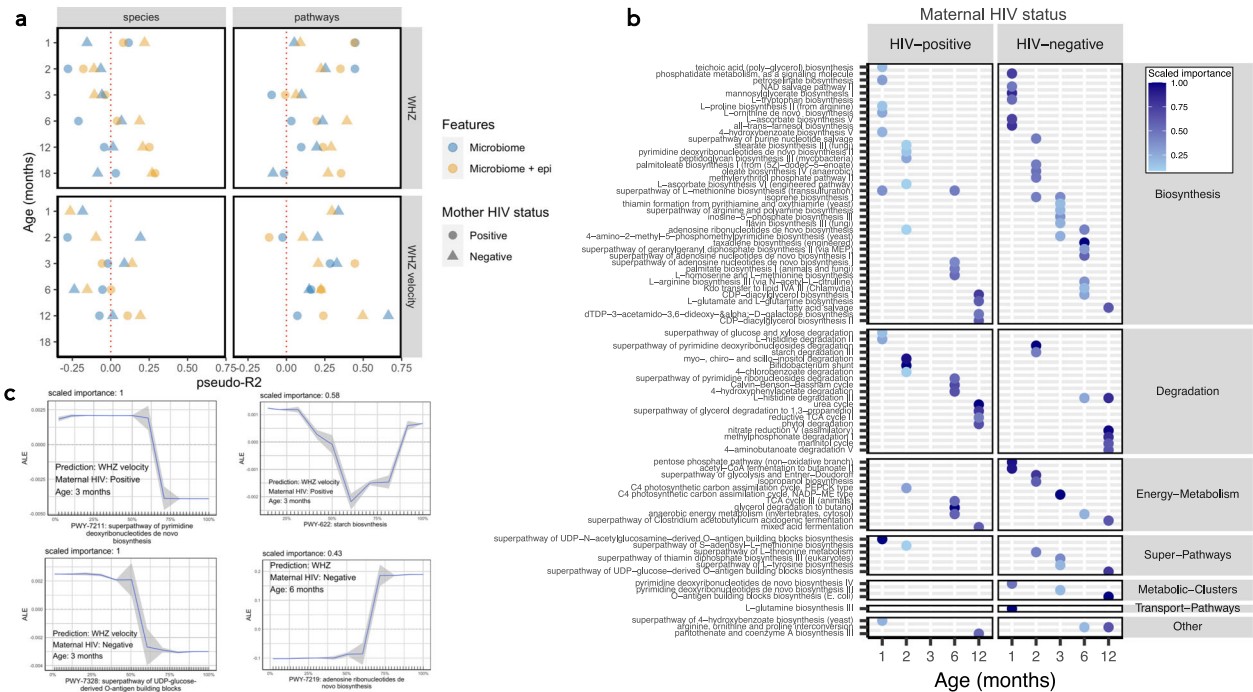

**Fig. 6 | Prediction of attained WHZ and WHZ velocity using XGBoost models.** Performance of XGBoost models as assessed by pseudo-$R^2$ values for prediction of attained WHZ and WHZ velocity (WHZ increase per day to next study visit) using species or metagenomic pathways, stratified by age category and maternal HIV status (**a**). Models were run using microbiome features alone (species or metagenomic pathways; blue points) and in combination with epidemiological variables (yellow points). The top ranked pathways predicting WHZ at each age category are plotted (**b**), stratified by maternal HIV status and coloured by scaled importance in the XGBoost model (darkness of blue shading indicates feature ranking in the model). Accumulated effect plots (ALE) of representative pathways ranking highly in XGBoost model predictions display change in predicted linear growth (WHZ or WHZ velocity) by percentile of the feature abundance distribution (**c**). Tick marks on the x-axis are a rug plot of individual feature abundance percentiles. ALEs were generated using the *ALEplot* package and were plotted using *ggplot2*. Standard deviations (sd) were calculated per increment in microbiome feature and were used to calculate and plot increment-wise 95% confidence intervals as the average change in the outcome ±1.96(sd/sqrt(n)), where *n* is the number of observed feature values, and sd is the standard deviation of the change in the outcome variable in an interval. *n* = 854 and *n* = 455 for models predicting WHZ and WHZ velocity respectively.

This study is strengthened by the inclusion of a large longitudinal cohort of rural sub-Saharan African mother-infant pairs enrolled during pregnancy, covering much of the first 1000 days of healthy infant development. These novel data thereby add to the literature of infant microbiome succession, which, to date, has largely focused on high-income, industrialized populations[17,20]. High levels of exclusive breastfeeding, randomized complementary feeding interventions, high prevalence of maternal HIV infection and extensive environmental metadata allowed for comprehensive examination of nutritional and environmental features defining non-industrialized infant microbiome succession and its association with child growth in a population with a high prevalence of stunting. These results, generated using gold-standard whole metagenome sequencing, suggest that the functional activity of the infant gut microbiome in early infancy could act as a novel target for nutritional interventions combating child undernutrition.

The SHINE trial found that the WASH intervention had no impact on linear growth, whilst IYCF improved growth by 0.16 LAZ in CHU and 0.26 in CHEU[39,42]. Furthermore, WASH had no impact on carriage of enteropathogens or diarrheal incidence[14]. Here, we show that the improved WASH and IYCF interventions also had little impact on the infant gut microbiome from 1–18 months of age. Our results support observations from other geographical settings, showing a structured, programmed assembly of the gut microbiome in healthy children who are born by vaginal delivery and exclusively breastfed[17,18,20,43]. This cohort adds to the literature showing that the gut microbiome in this rural sub-Saharan African population undergoes a similar assembly. These data suggest that this programmed microbial maturation is driven strongly by age, and is robust to changes in WASH and complementary feeding, as delivered in this trial, and that potential microbiome-mediated pathways affecting early-life growth occur independently of these specific interventions. Improvements in growth as a result of the IYCF intervention are not driven by the microbiome, as supported by previous reports showing that the gut microbiome does not mediate the effect of lipid-based IYCF nutrient supplements on child growth[44]. More intensive interventions that target WASH, microbial exposures, nutrient intake and microbiota-directed foods during the first 2 years of life may be required to modify this programmed trajectory of gut microbiome succession.

The data reported here complement previous research associating the gut microbiome with infant growth. The composition and maturity of the gut microbiota has been shown to be disturbed during severe acute malnutrition (SAM) and could be used to predict growth recovery[22]. More recently, an "ecogroup" of 15 bacterial taxa has been identified that exhibits consistent covariation, thereby representing microbiota maturation, throughout the first 2 years after birth across different geographical cohorts and which is predictive of ponderal growth[45]. However, little research has examined microbiome maturation in the context of stunting, chronic undernutrition or adequately nourished children from low-income settings. We report similar maturation of the early-life microbiome in this stunting cohort, driven by many of the same age-predictive taxa as previously reported, notably *Faecalibacterium prausnitzii*. We extend this to report functional maturation of the early-life gut microbiota and find that, in addition to amino acid and B-vitamin biosynthetic pathways, methanogenesis from acetate (METH-ACETATE-PWY) was the pathway most predictive of age, despite the apparent lack of methanogens. The

predictive strength of this pathway may reflect accumulation of acetogenic species, including *Blautia wexlerae*, that feed into reactions upstream of the METH-ACETATE-PWY. However, our results contrast with previous cross-sectional studies reporting an association between the taxonomic composition of the gut microbiome and stunting[34–37]. A previous study from sub-Saharan Africa (Afribiota) reported significant differences in the fecal microbiome of stunted and non-stunted children between 2–5 years of age, hypothesizing that decompartmentalization of the gastrointestinal tract and overgrowth of oropharyngeal taxa are associated with stunting[34,35]. We report here no association between the taxonomic composition of the gut microbiome and linear growth, however our study examined children at younger ages to that of the Afribiota cohort, suggesting that differences in the taxonomic composition of the gut microbiome mediating linear growth may only manifest later in childhood.

We identified a range of metagenomic pathways that predicted linear and ponderal growth through 18 months suggesting that the potential influence of an altered gut microbiome on child growth is dependent upon a number of interacting metagenomic pathways. This discrepancy in the ability of functional metagenomic features versus taxonomic features to predict growth may suggest that metagenomic pathways contributing to differences in early-life growth may be harboured across a number of functionally redundant species. Pathways encoding biosynthesis of B vitamins were consistently amongst the top predictive features in models predicting both attained and LAZ and WHZ growth velocity. Previous evidence supports the importance of B vitamins in early-life growth whereby maternal folic acid supplementation increases infant birthweight[46]. In infants, vitamin B12 status is predictive of both linear and ponderal growth[47], however the largest randomized trial of B12 supplementation on infant growth to date showed no effect[48,49]. The gut microbiome biosynthesizes and metabolises B vitamins, including cobalamin (B12) and folate (B9), at levels similar to dietary intake, and abundance of B vitamin-synthesizing genes in the infant gut microbiome differs by delivery mode, antibiotic exposure[50], exclusive breastfeeding practices and geographic location, where vitamin biosynthesis genes are greater in Western settings[43]. Greater relative abundance of B vitamin biosynthetic pathways such as thiazole, tetrahydrofolate and flavin biosynthesis in the maternal gut microbiome predicted greater birthweight and neonatal growth in this same cohort, whilst biotin biosynthesis predicted reduced birthweight[30]. The gut microbiome transferred from mother to infants may influence the metabolic capacity of the infant microbiome to biosynthesize essential nutrients and influence downstream growth pathways.

Purine and pyrimidine biosynthetic pathways consistently contributed to growth predictions across all age groups. In mothers from this same cohort, purine and pyrimidine salvage pathways were associated with increasing birthweight[30]. Meta-analyses have also found that dietary nucleotide supplementation in infants significantly increases head circumference and rate of weight gain[51], suggesting that microbiome-derived nucleotide metabolism may play an important role in, or be a marker of, nutritional status in early infancy. Indeed, microbial purine and pyrimidine biosynthesis is critical for survival of both pathogens and commensals[52] and therefore alterations in metagenomic nucleotide biosynthesis may disrupt metabolic activity of the gut microbiota. Furthermore, glycogen synthase was the pathway most predictive of birthweight in maternal microbiomes, whereby higher abundance predicted lower birthweight. Here, 2 related pathways (PWY-622 starch biosynthesis, GLYCOGENSYNTH-PWY glycogen biosynthesis I (from ADP-D-Glucose)) were also ranked as highly predictive of growth velocity. Glycogen synthesis occurs as a starvation response in bacteria which facilitates transition into a biofilm state suggesting that microbiome starvation or slowed microbial multiplication/proliferation responses are associated with infant growth as early as the first months after birth[53,54]. Collectively, we identified B

vitamin and nucleotide synthesis metagenomic pathways, amongst others, as predictive of child growth, suggesting that metabolic activity of the gut microbiome may influence early-life growth and act as a target for future dietary interventions.

An important observation, however, is that many of these pathways predicting growth, including B vitamin and purine/pyrimidine biosynthesis, were also predictive of age and hence microbiota maturation. There was a strong association between age and growth in the SHINE cohort, whereby LAZ declined steadily between 1–18 months of age. This highlights the difficulty in delineating the independent effect of the gut microbiome on growth during infancy, when the microbiome is concurrently undergoing age-related maturation, which is by far the strongest contributor to gut microbiome variability. We also found that the accuracy of models predicting growth increased with age, as observed by a reduced MAE in older compared with younger age groups (Supplementary Fig. 4). Reasons for the decreasing MAE are unclear; however, it is possible that the gut microbiome of 12–18 month old children is better at predicting growth than at 1–6 months of age, thereby producing less error in XGBoost models. Our own data (Supplementary Fig. 3c, d) and previous data[55] show far less inter-individual variability in 12–18 month gut microbiomes versus 1–6 months. This reduced inter-individual variability suggests that models of microbiome composition/function and predicted growth perform better at the later age groups. Although we attempted to account for age-related effects by examining samples within specified age categories, the microbiome-growth relationship observed here may be confounded by age. Previous studies, employing MAZ as a maturation index to account for age have demonstrated microbiome maturation is disturbed in acutely malnourished children[22] but is not associated with linear growth[56]. Conversely, our observations that functional microbiome characteristics moderately predict changes in linear growth add novel findings to this literature but need to be replicated in other large cohorts examining the functional maturation of the gut microbiome throughout early childhood in similar settings.

We report that maternal HIV infection had a significant impact on the infant microbiome between 1–18 months of age. We previously reported that CHEU have a 16% higher prevalence of stunting, 40% higher risk of infant mortality and poorer cognitive development compared with CHU[33,57]. The results presented here raise the intriguing possibility that altered succession and assembly of the infant gut microbiome may drive some of these poorer clinical outcomes in CHEU. These findings are in line with previous reports of disturbed gut microbiome composition in CHEU[28,32]. A number of factors may explain these differences. Firstly, mothers living with HIV receive both antiretroviral therapy (81% in the SHINE trial) and a broad-spectrum prophylactic antibiotic (cotrimoxazole), whilst CHEU also receive prophylactic cotrimoxazole from 6 weeks of age. Previous evidence suggests that cotrimoxazole may impact gut microbiome succession throughout childhood, increasing resistance gene diversity and the prevalence of particular pathobionts[58,59], which may explain some of the differences in microbiome maturity and diversity observed here. However, we found the largest differences in gut microbiome composition and function in samples from infants <6 weeks of age, prior to cotrimoxazole initiation, suggesting that these findings were independent of antibiotic prophylaxis and that CHEU may acquire an altered microbiome from their mothers. We saw relatively minor differences, however, in the gut microbiome of mothers living with HIV or without HIV in this same cohort[30]. Although there were significant differences in compositional beta diversity, the only species that differed in abundance was *Treponema berlinense*, which was significantly less abundant in mothers living with HIV. Secondly, exclusivity of breast-feeding (EBF) is one of the strongest determinants of infant gut microbiome composition, however there was no significant difference in EBF rates between CHEU and CHU in this cohort, suggesting that

exclusivity of breast-feeding was not responsible for these differences. Previous research has shown that the HMO content of breast milk differs between mothers living with and without HIV[28]. HMOs are the among the primary substrates for digestion by the infant gut microbiome thereby fundamentally determining gut microbiome composition. Indeed, we found that *Bifidobacteria* species, which are primary degraders of HMOs were significantly less abundant in CHEU, as were genes involved in amino acid biosynthesis. Previous in-depth profiling of infant immune development found that a lack of *Bifidobacteria* in infancy is associated with systemic inflammation and immune dysregulation[24], which are also observed in CHEU[60–62], suggesting that the lack of commensal *Bifidobacteria* may mediate some of the poor immune, growth and clinical outcomes observed in CHEU. Importantly, these microbiome differences between CHU and CHEU were not reversed by WASH and IYCF interventions, suggesting that novel microbiota-directed interventions that enhance *Bifidobacteria* colonization or prevent the over-diversification of the early-life gut microbiome are warranted for future clinical trials in CHEU.

There are several limitations to this analysis: (i) Machine-learning prediction models are at inherent risk of over-fitting and bias depending on cross-validation approaches and model complexity. Our conservative approach to model tuning and feature selection aimed to minimize overfitting. The age models, which included an out-of-sample ('healthy') test dataset, produced comparable performance metrics, making overfitting less likely. However, it remains possible that some degree of over-fitting occurred in growth models, since an out-of-sample test dataset was not used, thereby possibly affecting the strength of our reported gut microbiome-related growth predictions. (ii) the SHINE microbiome sub-study intentionally included more women living with HIV than the main SHINE trial (30% versus 15%), and the proportion of HIV-exposed infants varied by age group. This resulted in some small subgroups in some of our age categories by HIV exposure analyses, which likely resulted in unstable predictions in certain XGBoost models; (iii) a significant proportion of the sequencing reads included in our datasets were not annotatable (median 58.6%) using the specified bioinformatic pipelines (MetaPhlAn3 and HUMAnN3). This large abundance of unknown sequences is common in samples derived from non-Western populations[63] and leads to inferences solely being made from the assignable fraction, potentially missing important microbiota features that are predictive of infant growth but are currently not represented in databases. Kmer-based bioinformatic tools (e.g. Kraken) and marker-based tools (e.g. MetaPhlAn) each have their own advantages and disadvantages. Tools relying on metagenome assembly may help in identifying some of these unknown sequences in datasets from less-studied populations in the future; (iv) *Escherichia coli* was one of the most prevalent bacterial species across all age groups; however, MetaPhlAn3 cannot differentiate between *E. coli* pathotypes. Different *E. coli* pathotypes, such as enteropathogenic and enteroaggregative *E. coli* have been associated with intestinal pathology and EED; however, we previously reported in the same cohort that some of these pathotypes were not associated with growth[14]; (v) we attempted to account for the age-related confounding of the microbiome-growth relationship by predicting attained growth and growth velocity in discrete age groups, but residual confounding may still be present, influencing our ability to identify microbiome features independently associated with growth; (vi) data on infant antimicrobial use, aside from cotrimoxazole use in CHEU, was incomplete (but infrequent), limiting our ability to confidently assess this as a potential confounder in early-life gut microbiome maturation; (vii) differences in microbiome composition and function may also be driven by differences in intestinal microbial load, motility and biogeography, which were not assessed; and (viii) finally, we chose mother-infant pairs with the most complete sample collection during follow-up, in order to strengthen our inferences about development of the gut microbiome over time, but which may introduce selection bias. Baseline characteristics of the microbiome sub-cohort were largely similar to those of the larger trial, suggesting that the microbiome sub-study cohort studied was largely representative of the larger SHINE trial.

Collectively, these data suggest that HIV exposure shapes maturation of the infant gut microbiota, and that the functional composition of the infant gut microbiome is moderately predictive of infant growth in a population at high risk of stunting. Novel therapeutic approaches targeting the gut microbiome may mitigate the poor clinical outcomes that are observed in CHEU, a growing population of children in sub-Saharan Africa. By contrast, current WASH and IYCF interventions fail to impact the infant gut microbiome and therefore transformative WASH and microbiome-targeted dietary interventions may prove to be more successful approaches to target the microbial pathways mediating early-life growth.

## Methods

### SHINE trial design
The study design and methods for The Sanitation Hygiene Infant Nutrition Efficacy (SHINE) trial and for the corresponding microbiome analyses, have been reported previously[64,65]. Briefly, SHINE was a 2 × 2 cluster-randomized trial, conducted between 2012 and 2017, to determine the independent and combined effects of improved infant and young child feeding (IYCF) and WASH on child stunting and anaemia in two rural Zimbabwean districts, Chirumanzu and Shurugwi (NCT01824940). 5280 pregnant women were cluster-randomized to one of four interventions: WASH, IYCF, WASH + IYCF, and Standard of Care (SOC). The SOC interventions, included in all trial arms, comprised exclusive breastfeeding promotion for all infants up to 6 months and strengthened prevention of mother to child transmission (PMTCT) of HIV services. The household WASH intervention was initiated during pregnancy and was designed to reduce exposure to human and animal feces, including, at the household level: construction of a ventilated improved pit latrine, installation of two hand-washing stations plus monthly delivery of liquid soap and water chlorination solution, provision of a play space for the infant, and hygiene counselling. The IYCF intervention was designed to improve infant diets using a small-quantity lipid-based nutrient supplement (SQ-LNS), provided to the infant from 6–18 months, and educational interventions promoting the use of age-appropriate, locally available foods and dietary diversity. Lastly, a combined trial arm, WASH + IYCF, evaluated the effects of both improved WASH and infant nutrition. Due to the nature of the interventions, the study was unblinded. The primary outcomes (LAZ and haemoglobin concentrations at 18 months) were assessed by comparing length and age of children against standardized World Health Organization growth standards for LAZ and using a Hemocue hemoglobinometer, for haemoglobin concentrations.

Infants were followed up at study visits at 1, 3, 6, 12 and 18 months of age. Length and weight were measured at each infant visit, as described previously[39]. Length-for-age z scores (LAZ) and weight-for-height z scores (WHZ) were calculated from length and weight measurements at each visit according to WHO Child Growth Standards. Epidemiologic data for the infants was collected from the baseline and follow-up visits using trial questionnaires that included maternal anthropometry, birth outcomes, baseline household WASH facilities, household wealth, maternal education, religion, parity, household size, dietary diversity, changes in breastfeeding and complementary feeding practices, food security, 7-day and 3-month infant health status, and antimicrobial use.

### HIV testing
HIV testing was conducted on mothers at the baseline visit using a rapid test algorithm (Alere Determine HIV1/2 test, followed by INSTI

HIV-1/2 test if positive). Those testing positive for HIV had CD4 counts measured (Alere Pima Analyser) and referral to local clinics; women were encouraged to begin co-trimoxazole prophylaxis and ART, to exclusively breastfeed, and to attend clinic at 6 weeks postpartum for early infant diagnosis and infant co-trimoxazole prophylaxis. Women testing negative for HIV were offered retesting at 32 gestational weeks and 18 months postpartum. Children of mothers living with HIV were offered testing for HIV at each of the study visits. Those who tested positive were referred to local clinics for ART. HIV was diagnosed using DNA PCR on dried blood-spot samples or RNA PCR on plasma in samples collected prior to 18 months. In samples collected after 18 months, HIV was diagnosed by PCR or rapid test algorithm, depending on samples provided. Children born to women living with HIV and who tested negative at 18 months were classified as HIV-exposed uninfected (CHEU). Inconclusive or discordant results were re-tested; if no further samples were available or repeat testing was inconclusive, children were classified as HIV-unknown.

## Microbiome sub-study

All CHEU and a subgroup of CHU from the SHINE study were enrolled into an Environmental Enteric Dysfunction (EED) sub-study (n = 1,656 mother-child pairs); these infants underwent intensive biological specimen collection at 1, 3, 6, 12 and 18 months of age[65]. The EED sub-study was therefore enriched for mothers living with HIV, by design. Sample selection for inclusion into the current microbiome study was conducted to enhance longitudinal profiling of the mother and infants gut microbiota. Of the mother-infant pairs within the EED sub-study, those with least one maternal fecal specimen (of 2 possible) and at least 2 infant fecal specimens (of 5 possible) were included in the gut microbiome analyses. An additional 94 samples collected at the 1 and 3-month visits, that did not meet these criteria, but had microbiome sequencing data available from a separate study examining rotavirus vaccine immunogenicity in the SHINE trial[66], were also included in these analyses. Infant ages varied at each study visit due to the allowable window around the visit date for the larger SHINE trial. Therefore, for this microbiome study, stool samples were re-categorized into 6 age groups corresponding to important stages in infant microbiome development: "1 month" (0–6 weeks), "2 months" (7 weeks – <3 months), "3 months" (3–6 months), "6 months" (6–9 months), "12 months" (9–15 months), and "18 months" (15–20 months). Sample sizes in each age category are provided in Supplementary Table 1.

## Sample collection

Study visits were conducted by trained study nurses in participants' homes. Sterile stool collection tubes were provided to mothers, who collected stool samples from their infants on the morning of each study visit. Samples were placed in cool boxes immediately upon collection by study nurses and transported by motorbike to field laboratories where they were aliquoted and stored at −80 °C within 6 h of collection before subsequent transport to the central laboratory in Harare for long-term storage at −80 °C. An aliquot of each stool sample was shipped on dry ice by courier to the British Columbia Centre for Disease Control in Vancouver, Canada. A strict cold chain was maintained throughout transport, ensuring no freeze-thaw cycles occurred between sample collection and processing.

## Whole metagenome library preparation and sequencing

DNA was extracted from 100–200 mg of stool samples using the Qiagen DNeasy PowerSoil Kit as per the manufacturer's instructions. DNA quantity was assessed by fluorometry (Qubit) and quality confirmed by spectrophotometry (SimpliNano). 1 μg DNA was subsequently used as input for metagenomic sequencing library preparation using the Illumina TruSeq PCR-free library preparation protocol, using custom end-repair, adenylation and ligation enzyme

premixes (New England Biolabs). The concentration and size of constructed libraries were assessed by qPCR and by TapeStation (Agilent). DNA-free negative controls and positive controls (Zymo-BIOMICS D6300) were included in all DNA extraction and library preparation steps. Libraries were pooled in random batches of 48 samples including one negative control. A set of specimens were subject to replicate DNA extraction, library preparation and, sequencing to estimate the magnitude of technical variability and batch effects among samples. Whole metagenome sequencing was performed with 125-nucleotide paired-end reads using either the Illumina HiSeq 2500 or HiSeqX platforms at Canada's Michael Smith Genome Sciences Centre, Vancouver, Canada.

## Bioinformatics

Quality control and bioinformatic processing of raw sequencing data was conducted using the publicly available *KneadData* (https://huttenhower.sph.harvard.edu/kneaddata/). Sequenced reads were trimmed of adaptors and filtered to remove low-quality, short (<70% raw read length), and duplicate reads, as well as those of human, other animal or plant origin, with default settings. Species composition was determined by identifying clade-specific markers from reads using MetaPhlAn3 with default settings[67]. Relative abundance estimates were obtained from known assigned reads, and unknown read proportions were estimated from total, assigned and unassigned, reads. Percent human DNA was estimated from *KneadData* output, using the proportion of quality-filtered reads that align to the human genome. Given the smaller viral genome sizes, sequencing depth, and limitations of MetaPhlAn3 for virus identification, we did not include viruses in our current analyses. We applied a minimum threshold of >0.1% relative abundance and ≥5% prevalence for all detected species. Metabolic pathway composition was determined using HUMAnN3 with default settings against the UniRef90 database[67]. Pathway abundance estimates were normalized using reads per kilobase per million mapped reads (RPKM) and then re-normalized to relative abundance. We applied a minimum relative abundance threshold of $3 \times 10^{-7}\%$ and ≥5% prevalence for all metagenomic pathways.

## Statistical analysis

All data were analysed using R (v.4.0.5). Microbiome data were handled using the phyloseq package (v1.34.0). Alpha diversity metrics were calculated using the *vegan* package (v2.5.7). A permutation test, implemented using the *coin* package in R, was performed to compare alpha diversity metrics between unbalanced age categories. Beta-diversity was estimated using the Bray-Curtis dissimilarity index and analysed by permutation analysis of variance (PERMANOVA). Differential abundance analysis of species or functional pathways was assessed using multiple regression analyses using the *MaAsLin2* v1.5.1 package[68] and applying default arguments (significance thresholds −p < 0.05, q < 0.25). Four covariates were chosen for adjustment in diversity analyses, microbiome maturity analyses and *MaAsLin2* regression models and included age in days at stool sample collection, exclusive breastfeeding status (recorded at 3 months old), delivery mode (caesarean section versus other), and randomised trial arm. These covariates were chosen based on biological plausibility and previous evidence of their influence on gut microbiome composition in large birth cohorts[17]. Adjustment for multiple comparisons was performed using the Benjamini-Hochberg false discovery rate (FDR).

The SHINE trial did not observe an interaction between the randomized WASH and IYCF interventions and growth; therefore, randomised trial arms were combined into WASH versus non-WASH arms and IYCF versus non-IYCF arms for specified analyses. We restricted the IYCF analysis to the 6, 12 and 18 month visits, corresponding to the period during which supplemental infant feeding

was introduced (from 6 months of age). All children, regardless of HIV status or exposure status were included in the growth analyses (875 total stool samples). 16 samples had missing ages and were excluded from age prediction models. Stool samples collected from children classified as HIV-unknown (24 samples) or HIV-positive (4 samples) at 18 months were excluded from direct comparisons of CHEU vs CHU infants.

## XGBoost models

Relationships between the infant microbiome and age or growth (attained LAZ/WHZ and WHZ/LAZ velocity) were evaluated using extreme gradient boosting machines (XGBoost). XGBoost builds an optimized predictive model by creating an ensemble from a series of weakly predictive models. XGBoost is also non-parametric, can capture non-linear relationships, and can accommodate high-dimensional data[69]. The XGBoost models were developed using microbiome relative abundances (species or pathways) and all epidemiologic variables.

XGBoost model selection was performed in 3 stages[30]. In brief, the *BayesianOptimization* function of the *rBayesianOptimization* package was used with 10-fold cross-validation to select model hyperparameters by minimizing the mean squared error (MSE). This 3-stage hyperparameter tuning and model building was performed for two feature sets, one comprising microbiome features and a second comprising microbiome plus epidemiologic features; this was done to assess model performance and to examine the contribution of epidemiologic versus microbiome features. In stage one, the model dataset included either microbiome features alone or microbiome and epidemiologic features. Models with the lowest MSE (in the 5th percentile) were retained, and from these models the variables that contributed to the top 95% of variable importance by proportion were retained; this filtering was applied separately to the microbiome and epidemiologic features. In stage two, models were built with the retained features obtained from stage one, and *BayesianOptimization* was re-run as for stage one but using leave-one-out cross-validation. Models containing microbiome features alone, or together with epidemiologic features, were again filtered according to feature importance as for stage one. In stage three, all retained epidemiologic variables, microbiome features, and hyperparameters selected in stage two were used to fit our final models, using leave-one-out cross-validation to minimize the MSE. Separate models were built for attained LAZ and WHZ and growth velocity outcomes[13]. Growth velocity was defined as LAZ/WHZ units of change per day between the specified study visit and the subsequent study visit. We assessed microbiota composition and functional pathways separately. XGBoost models were fit using the H20.ai engine and *h2o* R package interface with the *XGBoost* package. XGBoost model performance was evaluated using pseudo-$R^2$ and mean absolute error (MAE). Pseudo-$R^2$ values <0 indicated that the prediction of the model was worse than the mean response. Scaled relative importance for each model feature was used to identify the twenty most informative variables for further interpretation, where the most important variable is ranked first, and the importance of subsequent variables are relative to the first variable. The marginal relationships between the twenty most important features and each growth outcome were visualized for interpretation[40] using accumulated local effects plots (ALE). ALE plots can be interpreted as showing a marginal effect, adjusted for all covariates retained in the final model, showing the expected change in the outcome variable per increment in a model feature. The resulting effect sizes are plotted cumulatively and centered about the average effect size[41]. ALEs were generated using the *ALEplot* package, modified to compute confidence intervals, and were plotted using *ggplot2*. Standard deviations were calculated per increment and were used to calculate and plot increment-wise 95% confidence intervals. The code for the implementation of XGBoost and the ALE plots is available at https://github.com/ThadEdens/shine-analysis.

## Microbiome age

We investigated microbiome maturation by building an age prediction model using XGBoost and microbiome features only (species or pathways). A combination of leave-one-out cross validation and cross validation out-of-fold predictions was used as an alternative to the train-test approach. We partitioned the infants and their corresponding datasets into three groups: (1) CHU with LAZ > −2 at 18 month of age, who contributed >1 dataset ("healthy training set"; $n = 265$ samples), (2) remaining CHU infants with LAZ > −2 at 18 month of age, who contributed a single metagenomic dataset ("healthy test set"; $n = 66$ samples); and (3) CHEU or children with LAZ ≤ −2 at 18 months of age ("unhealthy test set"; $n = 528$). 16 samples that were missing exact ages were excluded from the model. Age was log transformed as a response in the XGBoost model. We performed the same 3-stage tuning and model building procedure, as described above. We generated model performance metrics, including pseudo-$R^2$, mean absolute error (MAE) and mean squared error (MSE) for the three sets. Leave-one-out cross validation was used to build a model on the 'healthy training set' ($n = 265$ samples). Model performance for the 'healthy training set' was computed using the observed response and cross validation out-of-fold predictions. Model performance for the 'healthy' and 'unhealthy' test sets was computed using the observed responses and the final model predictions. We exponentiated the predicted log transformed ages and plotted these values against the observed age. The predicted age using these models is referred to as 'microbiota age' for the models trained using species and 'metagenome age' for models trained using pathways. Differences in microbiome age and metagenome age scores were assessed using multivariable linear regression. To account for variance of microbiota ages with respect to chronological age within the age range of each study visit, a microbiota for age Z-score (MAZ) and metagenome-for-age Z-score (MetAZ) was also created using the microbiome age and metagenome ages as previously described[22]. A Z-score was calculated to account for variation in ages within each study visit using the following formula:

(Microbiota age of child − median microbiota age of 'healthy' child at same study visit)/standard deviation of microbiota age of 'healthy' child at same study visit.

## Reporting summary

Further information on research design is available in the Nature Portfolio Reporting Summary linked to this article.

## Data availability

The raw metagenome sequencing data generated in this study have been deposited in the European Bioinformatics Database under accession code PRJEB51728. The UniRef90 database used to annotate metabolic pathways is publicly available. (https://www.uniprot.org/help/uniref). Epidemiologic data files and final processed and annotated metagenome sequencing data files (taxa and pathways) are available at https://doi.org/10.5281/zenodo.7471082.

## Code availability

The code for the implementation of XGBoost and the ALE plots is available at https://github.com/ThadEdens/shine-analysis.

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

## Acknowledgements

We thank all the mothers, babies, and their families who participated in the SHINE trial and all members of the SHINE trial team (all members listed here: https://doi.org/10.1093/cid/civ844). We particularly thank the leadership and staff of the Ministry of Health and Child Care in Chirumanzu and Shurugwi districts and Midlands Province (especially environmental health, nursing, and nutrition) for their roles in operationalization of the study procedures, the Ministry of Local Government officials in each district who supported and facilitated field operations, Phillipa Rambanepasi and her team for proficient management of all the finances, Virginia Sauramba for management of compliance issues, and the programme officers at the Gates Foundation and the Department for International Development, who enthusiastically worked with us over a long period to make SHINE happen. Bill & Melinda Gates Foundation (OPP1021542 and OPP1143707; J.H.H. and A.J.P.), with a subcontract to the University of British Columbia (20R25498; A.R.M.). United Kingdom Department for International Development (DFID/UKAID; J.H.H. and A.J.P.). Wellcome Trust (093768/Z/10/Z, 108065/Z/15/Z, 206455/Z/17/Z, 203905/Z/16/Z and 210807/Z/18/Z; A.J.P., R.C.R. and C.E.). Swiss Agency for Development and Cooperation (J.H.H. and A.J.P.). US National Institutes of Health (2R01HD060338-06; J.H.H.). UNICEF (PCA-2017-0002; J.H.H. and A.J.P.). The funders had no role in the design of the study and collection, analysis, and interpretation of data and in writing the manuscript.

## Author contributions

A.R.M., L.E.S., R.J.S., J.H.H. and A.J.P. conceptualized and designed the study. K.M., R.N., B.C., F.D.M., N.V.T., J.T., and B.M. collected data and biospecimens. H.M.G., I.B., S.K.G., R.C.R., F.F. and L.C. processed biospecimens. T.J.E. conducted bioinformatics and machine learning analysis. R.C.R., A.R.M., T.J.E., L.C., C.E. and E.K.G. analysed and interpreted the data. R.C.R., A.R.M., L.C. and T.J.E. wrote the original manuscript draft. All authors reviewed the manuscript. A.R.M., A.J.P. and J.H.H. supervised and verified the data.

## Competing interests

TJE was paid a scientific consulting fee in relation to the analysis of the data presented here by Zvitambo Institute for Maternal and Child Health Research. RCR declares remittance from Abbott Nutrition Health Institute (March 2022) and Nutricia (May 2021) for public conference talks outside the submitted work. All other authors declare that they have no competing interests.

## Ethics approvals

All SHINE mothers provided written informed consent. The Medical Research Council of Zimbabwe (MRCZ/A/1675), Johns Hopkins Bloomberg School of Public Health (JHU IRB # 4205.), and the University of British Columbia Ethics Board (H15-03074) approved the study protocol, including the microbiome analyses. The SHINE trial is registered at ClinicalTrials.gov (NCT01824940).

## Additional information

[1]Blizard Institute, Queen Mary University of London, London, UK. [2]Devil's Staircase Consulting, West Vancouver, BC, Canada. [3]Department of Microbiology and Immunology, University of British Columbia, Vancouver, BC, Canada. [4]Zvitambo Institute for Maternal and Child Health Research, Harare, Zimbabwe. [5]Department of International Health, Johns Hopkins Bloomberg School of Public Health, Baltimore, MD, USA. [6]School of Population and Public Health, University of British Columbia, Vancouver, BC, Canada. [7]Department of Public and Ecosystem Health, Cornell University, Ithaca, NY, USA. [8]Department of Experimental Medicine, University of British Columbia, Vancouver, BC, Canada. [9]Goshen College, Goshen, Indiana, IN, USA. [10]British Columbia Centre for Disease Control, Vancouver, BC, Canada. [11]Present address: Microenvironment & Immunity Unit, INSERM U1224, Institut Pasteur, 75015 Paris, France. [12]These authors contributed equally: Ruairi C. Robertson, Thaddeus J. Edens, Lynnea Carr. ✉e-mail: amee.manges@ubc.ca

