## [Peer Review File · Nature Communications]

Reviewer comments, first round review –

Reviewer #1 (Remarks to the Author):

This study investigated the correlation between the gut microbiome and early-life growth in a population with high prevalence of stunting. The authors explored the gut microbiome in 335 children from rural Zimbabwe from 1-18 months of age. They found that the early-life gut microbiome undergoes programmed assembly that is unresponsive to randomized sanitation and nutrition interventions intended to improve linear growth. They also evaluated the effect of maternal HIV infection on the early-life microbiome. This study provides some novel insights into the gut microbiome associated with early-life growth restriction, and particularly it proposes a machine learning model to predict both attained linear and ponderal growth and growth velocity. I'd like to recommend it to be accepted if the authors could clearly clarify the novelty of this study and address the following concerns.

1. Recently, several large-scale infant microbiome studies have been published. Unfortunately, this study neglected all these relevant publications. For example, Xiao et al. investigated 13,776 fecal samples from 1956 infants between 1 and 3 years of age, based on multi-population cohorts covering 17 countries (PMID: 34429130); Beller et al. investigated the microbiota over the first year of life in eight densely sampled infants (PMID: 34903050). In addition, there are also several neonatal microbiome studies associated with growth restriction (PMID: 35387876) or malnutrition (PMID: 33826814). The authors should discuss them in context and emphasize the novel findings of this study.
2. As the authors mentioned (page 6, lines 12-13, and page 23, lines 1-4), a significant proportion of the sequencing reads cannot be annotated (median 58.6%), which may make a great difference to the results. Other pipelines (e.g., Kraken) or databases (e.g., NR) should be considered.
3. When the comparison between HIV exposure groups was made, how did the authors control other covariates (e.g., WASH or IYCF intervention), please clarify.
4. Considering the sample size is imbalanced between CHEU and CHU groups (91 vs. 27), an additional permutation test is suggested to be used to reduce the bias.
5. Figure 4c-h, it seemed that the results for the pathway were not significant (page 11, line 12-22; page 12, line 1-2); if so, please give the p value or q value.
6. Description of subfigure c of figure 5 is missing in the figure legend, page 56; so is figure 6, page 58
7. Wrong caption in figure 6, LAZ -> WHZ, page 58

Reviewer #2 (Remarks to the Author):

In this study the gut microbiome of 335 children aged 1-18 months of age from regions of rural Zimbabwe were analysed. Microbiome maturation was assessed along with the impact of randomized WASH and nutrition interventions and associations with environmental exposures including maternal HIV infection. Stool samples were categorized into 6 age groups corresponding to important stages in infant microbiome development: "1 month", "2 months" 3 months, 6 months 12 and 18 months.

Methodology included compositional and functional metagenomic analysis together with

epidemiological approaches, and machine learning to test the ability of the early-life gut microbiome to predict both attained linear and ponderal growth and growth velocity through the first 18 months of life.

The noteworthy results are that early-life gut microbiome undergoes programmed assembly that is unresponsive to randomized sanitation and nutrition interventions intended to improve linear growth. It was found that HIV exposure shapes maturation of the infant gut microbiota, with HIV infection being associated with over-diversification and over-maturity of the early-life gut microbiome in addition to reduced abundance of Bifidobacterium species. Using machine learning models (XGBoost), taxonomic microbiome features are poorly predictive of growth however functional metagenomic features, particularly B-vitamin and nucleotide biosynthesis pathways, moderately predict both attained linear and ponderal growth and growth velocity.

Specific Comments

The paper is well written however the authors should emphasize much more it's innovation as there are several paper looking at time series of babies.

There is not enough detail provided in the methods for the work to be reproduced. The paper suggests that some infants have more than one sample, but no further detail is given. It is difficult to assess the appropriateness of all of the performed analyses without better understanding the longitudinal nature of the data.

It should be clarified as to how many samples were analysed at each time point. Furthermore, some of the details of the cohort are not clearly described, such as how many of children have 1, 2 and 3 gut microbiota samples, the details of the collected exposure data, etc. A table with cohort characteristics for each time should be included.

Detailed nutritional data throughout the study period, including milk quality during breast feeding and on supplemental infant feeding introduced from 6 months of age should be included. Without it, it is not possible to draw conclusions about the impact of specific dietary interventions on the developing gut microbiome from compositional and functional perspectives. There is a huge amount of temporal and inter-individual variation in early life in the gut microbiota, particularly with a wide range of perinatal exposures.

The writing style is somewhat technical and difficult to follow. This article could not be easily understood by someone outside of the field of microbiome research, and the paper sometimes reads like a summary of results from all of the possible analyses in a microbiome software package.

The discussion does not clearly convince me of the importance or impact of the results.

A concern with the work presented is that it seems that much of the data is not significant after corrections which I think is a problem. All results that are not significant should be removed.

No mention is made of antibiotic exposure during the study period. This is important information, particularly given the evidence that numerous rounds of antibiotics can have a compounding effect.

Was any other medication for HIV given to the mothers during pregnancy and while breast feeding?

Minor

p. 1 LINE 11 Bifidobacterium species

P. 2 Line 4 Stunting, or linear growth failure, is a form of chronic undernutrition is not correct, as actually Stunting, arises as a result of chronic undernutrition

Reviewer #3 (Remarks to the Author):

The authors analyse the gut microbiome maturation of 335 children from rural Zimbabwe aged between 1 and 18 months (n samples = 875). In particular, the study aims at exploring the potential associations between nutrition interventions, sanitation and hygiene practices, and maternal HIV infection on the gut microbiome assembly of HIV-exposed but uninfected (CHEU) children, a population characterized by high rate of stunting.

The gut microbiome of children born to HIV+ mothers was characterized by both compositional and functional over-maturity compared to the one of children born from HIV- mothers. Sanitation and hygiene practices as well as nutrition interventions showed little-to-no impact on the gut microbial maturation. No significant association was found between the gut microbiome composition and growth scores, while microbiome functionality was only moderately predictive.

Major Concern:

Overall, this study adds moderate value to the existing literature on microbiome association with stunting. While the presented (largely negative) results are of general interest to the field, they are presented in an excessively descriptive way, whilst providing very limited in-depth discussion into their biological implications. I therefore encourage the authors to expand on the biological significance of their results (such as i) the positive versus negative impact of the gut microbiome over-maturity on health, ii) resilience of the gut microbiome to changes in sanitation and hygiene practices as well as iii) the moderate association of growth scores with microbial functionality but not composition).

Minor Concerns:

- Throughout the text the authors refer to "HIV-unexposed (CHU)" and "HIV-exposed but uninfected (CHEU)" groups and then repeatedly switch to the equivalent but different terminology "children from HIV-positive/negative mothers".
- The acronyms CHU/CHEU are very similar to each other and are unintuitive. For example, CHU (children HIV exposed) should instead be HUC (HIV exposed children). A much more readable alternative that would differentiate well the two groups is HIVExp- and HIVExp+ Inf-
- While it is mentioned in the Discussion that cotrimoxazole treatment might impact the results >6 weeks, the authors should expand on the potential impact of this broad-spectrum antibiotic on the gut microbiome assembly and maturation.
- Figure 5 and 6 have identical caption title.
- Figure 5 and 6: specify that "HIV-positive" and "HIV-negative" refer to the maternal status.
- Page 3, Line 7: "In addition to research investigating"
- LAZ and WHZ acronyms are described in the Methods (page 26) but are mentioned as early as page 6. Add at the first reference a pointer to the detailed description and specify that the acronyms are scores.
- Page 15, Line 20 and Page 16, line 14: "through the first 18 months after birth" implies that sampling was performed during the whole duration of the 18 months (starting from birth), which is not the case.

Reviewer #1 (Remarks to the Author):

This study investigated the correlation between the gut microbiome and early-life growth in a population with high prevalence of stunting. The authors explored the gut microbiome in 335 children from rural Zimbabwe from 1-18 months of age. They found that the early-life gut microbiome undergoes programmed assembly that is unresponsive to randomized sanitation and nutrition interventions intended to improve linear growth. They also evaluated the effect of maternal HIV infection on the early-life microbiome. This study provides some novel insights into the gut microbiome associated with early-life growth restriction, and particularly it proposes a machine learning model to predict both attained linear and ponderal growth and growth velocity. I'd like to recommend it to be accepted if the authors could clearly clarify the novelty of this study and address the following concerns.

1. Recently, several large-scale infant microbiome studies have been published. Unfortunately, this study neglected all these relevant publications. For example, Xiao et al. investigated 13,776 fecal samples from 1956 infants between 1 and 3 years of age, based on multi-population cohorts covering 17 countries (PMID: 34429130); Beller et al. investigated the microbiota over the first year of life in eight densely sampled infants (PMID: 34903050). In addition, there are also several neonatal microbiome studies associated with growth restriction (PMID: 35387876) or malnutrition (PMID: 33826814). The authors should discuss them in context and emphasize the novel findings of this study.

The authors acknowledge and appreciate the papers highlighted by Reviewer 1 that were overlooked in the manuscript. The authors had cited Chen et al (PMID: 33826814), however acknowledge that the results from Chen et al and other recent infant microbiome papers should be discussed more and put into context. The authors have now addressed this point by citing these papers in the introduction (Page 3, Lines 17-20) and by providing context in the discussion (Page 17-18).

2. As the authors mentioned (page 6, lines 12-13, and page 23, lines 1-4), a significant proportion of the sequencing reads cannot be annotated (median 58.6%), which may make a great difference to the results. Other pipelines (e.g., Kraken) or databases (e.g., NR) should be considered.

The authors appreciate this important point. A number of other pipelines and databases are available for the analysis of metagenomic data, including those cited by the Reviewer, each with their advantages and disadvantages. Early on in the project, the authors examined both Kraken and MetaPhlAn pipelines for taxonomic profiling. Using our data, Kraken2 identified a much larger number of unique taxa, compared to MetaPhlAn2. However, the percent unknown or unassigned was high in results employing both pipelines; therefore, an alternative pipeline would not necessarily resolve the issue raised by the Reviewer. In benchmarking studies, MetaPhlAn has been associated with higher precision (PMID: 28967888), while Kraken2 has been shown to have a higher sensitivity (recall), but also a higher proportion of false positives (PMID: 31398336). Furthermore, Kraken2 does not account for genome size in relative abundance estimates, which can result in some biases (PMID: 33986544), while MetaPhlAn (a marker-based tool), is not sensitive to differences in genome length. While no method is perfect, the authors elected to use MetaPhlAn2 (and subsequently MetaPhlAn3) to minimize the number of false positives, especially for those taxa at relatively low abundances; this was important for analyses of gut metagenomes, which are highly diverse and are expected to have large numbers of low abundance taxa, particularly in non-industrialized settings. In addition, MetaPhlAn has been consistently used in other large infant microbiome profiling studies (PMID: 30356183) including those cited by the Reviewer (PMID: 34429130) thereby allowing the most suitable comparison with the current literature. The authors have added text to the 'limitations' paragraph of the discussion to discuss this (Page 25, Lines 2-6).

3. When the comparison between HIV exposure groups was made, how did the authors control other covariates (e.g., WASH or IYCF intervention), please clarify.

Four covariates were included in multivariable regression analyses (implemented using MaAsLin2) comparing differential abundance of species or genes between HIV exposure

groups: exact age at stool sample collection (measured in days from birth), exclusive breastfeeding status (measured at 3 months of age; binary variable), delivery mode (C-section vs other; binary variable), and randomised trial arm (WASH, IYCF, WASH+IYCF, standard of care). These covariates were chosen based on biological plausibility and previous evidence of their influence on gut microbiome composition in other large birth cohorts. Alpha diversity, beta diversity and microbiome maturity were initially assessed using univariable analysis. The authors have included these covariates in PERMANOVA analyses and multivariate regression analyses of diversity/microbiome maturity scores, which has led to minor changes in p-values and effect sizes, but the overall results and inferences remain unchanged. In the original manuscript, this adjustment approach is described in the results section on Page 11, Lines 20-21 and in the methods section, on Page 31, Lines 23 and Page 32, Lines 1-3. The authors have made minor amendments to this statement to clarify that these covariates were adjusted for in the models.

4. Considering the sample size is imbalanced between CHEU and CHU groups (91 vs. 27), an additional permutation test is suggested to be used to reduce the bias.

We have performed a permutation test to compare the alpha diversity metrics (Shannon, evenness and richness) for CHEU and CHU infants at the 12-month visit, accounting for the imbalanced sample size at this time-point. Using the *coin* package in R, the p-values for the alpha diversity metric comparison remain significant (Shannon, p-value = 0.0012; Evenness, p-value = 0.017; Richness, p-value = 0.005). The permutation results have been added to Page 10, Lines 16-19 and a description of the permutation implementation has been added to the methods section.

The authors acknowledge the lack of clarity regarding sample size as mentioned by both Reviewer 1 and Reviewer 2. The sample sizes mentioned above (91 vs 27) are solely for samples within the "12-month" age bracket of the CHEU vs CHU analysis. This microbiome sub-study was embedded within the larger SHINE trial. Due to the lack of complete longitudinal sampling at every time point for every infant, sample sizes differed at some time-points. To clarify this, the authors have added an additional table to the supplementary data detailing the number of samples analysed at each time-point (Table S1) within each age-bracket, split by maternal HIV status. These sample sizes pertain to all descriptive analyses of microbiome composition, analyses of randomized trial arm effects and XGBoost models assessing growth. A small number of children had a positive HIV test result at 18 months of age and were excluded, or an unknown HIV test, meaning they could not be assigned confidently as CHEU or CHU; these children were also excluded from the analyses of direct comparisons between CHEU and CHU (Figures 3 and 4).

5. Figure 4c-h, it seemed that the results for the pathway were not significant (page 11, line 12-22; page 12, line 1-2); if so, please give the p value or q value.

The authors acknowledge the lack of clarification on p and q values for these specified plots. Each of the results presented in Figs. 4c-h generated p values <0.05 and q values <0.25 in multivariate regression analyses using MaAsLin2. These thresholds for "significance" are the default values applied by the multivariate regression model (<https://huttenhower.sph.harvard.edu/maaslin/>). To aid in the interpretation of these results, the authors have also included the model coefficients in the plots. The full results of these models are also provided in Table S4. The authors have also made minor edits to the figure legend to clarify this.

6. Description of subfigure c of figure 5 is missing in the figure legend, page 56; so is figure 6, page 58

Thank you for highlighting this. Panel c in both figures represents ALE plots of representative pathways ranking highly in XGBoost model predictions. Descriptions of these ALE plots are included in the figure captions, which now indicate reference to panel c.

7. Wrong caption in figure 6, LAZ -> WHZ, page 58

Thank you for spotting this. The authors have now amended this error in the revised manuscript

Reviewer #2 (Remarks to the Author):

In this study the gut microbiome of 335 children aged 1-18 months of age from regions of rural Zimbabwe were analysed. Microbiome maturation was assessed along with the impact of randomized WASH and nutrition interventions and associations with environmental exposures including maternal HIV infection. Stool samples were categorized into 6 age groups corresponding to important stages in infant microbiome development: "1 month", "2 months" 3 months, 6 months 12 and 18 months.

Methodology included compositional and functional metagenomic analysis together with epidemiological approaches, and machine learning to test the ability of the early-life gut microbiome to predict both attained linear and ponderal growth and growth velocity through the first 18 months of life.

The noteworthy results are that early-life gut microbiome undergoes programmed assembly that is unresponsive to randomized sanitation and nutrition interventions intended to improve linear growth. It was found that HIV exposure shapes maturation of the infant gut microbiota, with HIV infection being associated with over-diversification and over-maturity of the early-life gut microbiome in addition to reduced abundance of Bifidobacterium species. Using machine learning models (XGBoost), taxonomic microbiome features are poorly predictive of growth however functional metagenomic features, particularly B-vitamin and nucleotide biosynthesis pathways, moderately predict both attained linear and ponderal growth and growth velocity.

Specific Comments

1. The paper is well written however the authors should emphasize much more it's innovation as there are several paper looking at time series of babies.

The authors have revised the introduction and discussion to further emphasize the novelty of this study. Study results stem from a longitudinal cohort comprised of rural African mother-infant pairs enrolled during pregnancy, which covers most of the first 1000 days of infant development. The cohort enrolled a population with a high HIV antenatal prevalence (15%), enabling the authors to examine the impact of HIV exposure during gestation and early life on the infant microbiota. High levels of exclusive breastfeeding and similar complementary feeds between infant groups reduced confounding due to dietary differences, and a comprehensive epidemiologic and clinical data are available. Finally, the cohort was subject to randomized interventions which were expected to directly influence the infant microbiota, and microbiota compositional and functional changes were captured using gold standard whole metagenome sequencing.

2. There is not enough detail provided in the methods for the work to be reproduced. The paper suggests that some infants have more than one sample, but no further detail is given. It is difficult to assess the appropriateness of all of the performed analyses without better understanding the longitudinal nature of the data.

The analyses can be grouped into two parts. The first part involves the detailed characterization of the alpha and beta diversity metrics by infant age, and according to important cohort characteristics such as infant HIV exposure status and SHINE intervention arm. These analyses were performed using standard methods to contrast diversity metrics (phyloseq and vegan packages). The authors also performed multivariable analyses using MaAsLin2 to identify uniquely enriched or depleted taxa associated with the same SHINE cohort characteristics. MaAsLin2 was performed using the default parameters, as described in the methods section. A new supplemental table (Table S2) has been added which summarizes the number of study subjects included in the analyses by age group and HIV exposure status.

The second part of our analyses focused on using microbiome features to predict infant age and growth using XGBoost, a machine learning approach. These methods are outlined in detail in the methods and the authors have now added a link to a github site containing the code to implement the XGBoost models [<https://github.com/ThadEdens/shine-analysis>]. The inclusion of our R code, including readme and documentation, will add clarity to our methods and will facilitate the reproducibility of our analyses by other groups.

3. It should be clarified as to how many samples were analysed at each time point. Furthermore, some of the details of the cohort are not clearly described, such as how many of children have 1, 2 and 3 gut microbiota samples, the details of the collected exposure data, etc. A table with cohort characteristics for each time should be included.

The authors acknowledge the lack of clarity in the sample number breakdown in each age category. The authors have now added an additional table in the supplementary data (Table S1) to detail the number of samples in each age category. The baseline characteristics of the infants in the microbiome sub-study are included in Table S2. A majority of these are fixed baseline characteristics which would not change over the follow-up period. The distribution of fecal specimens collected by age (the biggest driver of microbiome changes) is provided in Figure S1 and the mean number of samples per child (2.6 ± 1.3) is cited in the text.

4. Detailed nutritional data throughout the study period, including milk quality during breast feeding and on supplemental infant feeding introduced from 6 months of age should be included. Without it, it is not possible to draw conclusions about the impact of specific dietary interventions on the developing gut microbiome from compositional and functional perspectives. There is a huge amount of temporal and inter-individual variation in early life in the gut microbiota, particularly with a wide range of perinatal exposures.

The authors agree with Reviewer that breastmilk quality, breastfeeding patterns, and complementary feeds are critically important to microbiome maturation and infant growth. However, given the complexity of data collection and specimen analyses required for the SHINE Trial and the microbiome sub-study it was not feasible to also request breast milk samples for analysis from participating mothers, nor include detailed measurements of infant weight before and after feeding. A 2019 SHINE Team paper reported that the SHINE community health worker behaviour-change interventions strongly and positively influenced the early initiation and exclusivity of breastfeeding (PMID: 30937421). Overall, SHINE produced a sustained level of exclusive breastfeeding of 90% at 4 months of age and 75% at 6 months of age. The authors feel this reduced the heterogeneity in breastfeeding and dietary exposures among the infants included in our study. The authors also point to the fact that detailed information on complementary feeds was captured for all of the infants. All of these dietary variables were offered to the XGboost growth prediction models and none were identified as important for growth, although we did not directly examine the relationship between diet and infant microbiome composition or function. Moreover, as described in the manuscript there were no differences in the microbiome between infants in the IYCF versus non-IYCF arms. Finally, in all diversity and compositional analyses, exclusive breastfeeding status (measured at 3 months of age) was incorporated as a covariate into the analyses. We therefore believe that, with the available data, we have, to the best of our ability, accounted for differences in breastfeeding status and complementary feeding.

5. The writing style is somewhat technical and difficult to follow. This article could not be easily understood by someone outside of the field of microbiome research, and the paper sometimes reads like a summary of results from all of the possible analyses in a microbiome software package.

The authors acknowledge the Reviewer's comment regarding the technical language included in the manuscript. The authors have added sentences in the results and discussion that provide simple summaries of the key findings. The study analyses were extensive, the authors were simultaneously investigating infant compositional and functional microbiome maturation over five study visits (over 18 months of life), in children who were HIV-exposed during gestation or not, and subject to the WASH and

IYCF SHINE interventions, and also examining how the microbiome influences linear and ponderal growth, while accounting for key epidemiologic variables known to influence the intestinal microbiota. The authors elected to present the SHINE infant microbiome maturation story, in all its complexity, rather than break up the analyses into separate manuscripts. The authors acknowledge that this decision makes the manuscript denser, but feel that this is balanced by providing a comprehensive picture of the SHINE infant microbiome. We are hopeful that the new summaries at the end of each section and the amendments to the results and discussion will help readers navigate the key findings.

6. The discussion does not clearly convince me of the importance or impact of the results.

In response to point #1 above, the authors have revised the discussion to highlight the importance and impact of the study results, while not overstating or over-interpreting the results.

7. A concern with the work presented is that it seems that much of the data is not significant after corrections which I think is a problem. All results that are not significant should be removed.

The authors respectfully disagree with the opinion of the Reviewer that all non-significant results should be removed. First, our study produced a number of significant results. The authors identified comprehensive differences in gut microbiome composition, function, diversity and maturity between CHEU and CHU infants, particularly in early life, and our data show that microbiome functions can predict both attained and future growth, albeit moderately.

Second, our study produced a number of noteworthy, non-significant results which we believe are important to share, to avoid bias in interpretation of our data. The authors showed that long-term WASH (building latrines, hand-washing stations, chlorinated water, etc.) and providing a daily nutritional supplement (Nutributter) from 6-12 months of age had little to no effect on the acquisition and assembly of the fecal microbiome. The authors believe that these negative results are important when put into context of the outcomes of the larger SHINE intervention trial, which showed that the WASH strategies typically deployed in rural areas of low-income countries do not reduce diarrhoea or improve growth in young children. These findings have been instrumental in changing the research and programme landscape of WASH in global health programs which will have important consequences on strategies to combat child stunting.

Until recently, microbiome research has relied largely on taxonomic analysis. Here, the authors show that microbiome taxonomy is insufficient, whilst metagenomic functions are sufficient, to modestly predict child growth. Previous data suggest that taxonomic composition of the gut microbiota can predict growth, albeit in the context of growth recovery following severe acute malnutrition, rather than normal, age-related ponderal growth. These data show that the same approach cannot be used for linear growth. The authors believe that in the context of other infant microbiome studies these results are noteworthy and important to report. Moreover, the authors expected there to be taxonomic microbiota differences in response to the IYCF SHINE intervention, but were surprised that there were minimal compositional differences. Diet is one of the most important determinants of microbiota composition and function; therefore, the very modest differences the authors report in IYCF versus non-IYCF exposed infants will be important to future nutritional interventions designed to modify the microbiota or improve child health.

8. No mention is made of antibiotic exposure during the study period. This is important information, particularly given the evidence that numerous rounds of antibiotics can have a compounding effect.

The authors acknowledge the critical role that antibiotics play in early-life gut microbiome assembly. Data on antibiotic usage were recorded by maternal recall (14-day recall), which has the potential for mis-identification of medication. However, these data were included in the XGBoost models of infant growth, along with all other epidemiologic data, but none emerged as important to growth prediction. Only 2-8% of HIV-negative mothers

reported infant antibiotic use across visits, compared with 55-76% of HIV-positive mothers, which was largely attributed to infant cotrimoxazole use as part of WHO guidelines for CHEU. Figures for antibiotic use are now cited in the results (Page 6, Lines 6-9 and Page 10, Lines 11-14). The authors have discussed this limitation in the discussion (Page 25, Lines 14-16).

9. Was any other medication for HIV given to the mothers during pregnancy and while breast feeding?

Mothers living with HIV also received ART during pregnancy and breastfeeding. In brief, HIV testing was conducted by the SHINE trial team and mothers testing positive were referred to local clinics. Women were encouraged to initiate co-trimoxazole and ART, to exclusively breastfeed, and to attend clinic at 6 weeks postpartum for early infant diagnosis and co-trimoxazole. The trial did not provide HIV care or dispense ART, but SHINE did encourage attendance at local clinics, where prevention of mother to child transmission (PMTCT) coverage in the district was high but not universal. Maternal ART was documented based on maternal report and review of medical records. 81% of HIV-positive mothers had documented ART use during pregnancy. The authors have added a line in the discussion to include this information (Page 23, Line 6-12).

Minor

10. p. 1 LINE 11 Bifidobacterium species

This has been amended in the text

11. P. 2 Line 4 Stunting, or linear growth failure, is a form of chronic undernutrition is not correct, as actually Stunting, arises as a result of chronic undernutrition

This has been amended in the text to provide a more nuanced description:

"Stunting, or linear growth failure, arises from a network of underlying factors including inadequate dietary quantity and quality"

Reviewer #3 (Remarks to the Author):

The authors analyse the gut microbiome maturation of 335 children from rural Zimbabwe aged between 1 and 18 months (n samples = 875). In particular, the study aims at exploring the potential associations between nutrition interventions, sanitation and hygiene practices, and maternal HIV infection on the gut microbiome assembly of HIV-exposed but uninfected (CHEU) children, a population characterized by high rate of stunting.

The gut microbiome of children born to HIV+ mothers was characterized by both compositional and functional over-maturity compared to the one of children born from HIV- mothers. Sanitation and hygiene practices as well as nutrition interventions showed little-to-no impact on the gut microbial maturation. No significant association was found between the gut microbiome composition and growth scores, while microbiome functionality was only moderately predictive.

Major Concern:

Overall, this study adds moderate value to the existing literature on microbiome association with stunting. While the presented (largely negative) results are of general interest to the field, they are presented in an excessively descriptive way, whilst providing very limited in-depth discussion into their biological implications. I therefore encourage the authors to expand on the biological significance of their results (such as i) the positive versus negative impact of the gut microbiome over-maturity on health, ii) resilience of the gut microbiome to changes in sanitation and hygiene practices as well as iii) the moderate association of growth scores with microbial functionality but not composition).

The comments by Reviewer 3 echo comments from the other Reviewers regarding the contextualization of the SHINE microbiome results with the literature and drawing out the key findings and interpretation. The authors have revised the discussion to address these concerns regarding interpretation of the novelty of these results.

Minor Concerns:

Throughout the text the authors refer to "HIV-unexposed (CHU)" and "HIV-exposed but uninfected (CHEU)" groups and then repeatedly switch to the equivalent but different terminology "children from HIV-positive/negative mothers".

The authors acknowledge this confusion and aim to clarify the terms here. For a majority of these analyses (microbiome characterization, trial arm comparisons, XGBoost growth models), participants were separated by maternal HIV status. In the larger SHINE trial, a known maternal HIV status was an eligibility criterion for inclusion. Therefore, for these separate analyses, children are referred to as (children born to HIV-negative mothers vs children born to HIV-positive mothers).

As detailed in the methods section (Page 28), children born to HIV-positive mothers were offered HIV testing at each study visit, however, a known HIV status was not an inclusion criterion for children. Therefore, a small number of samples (n=24) came from children with an "unknown" HIV status at 18 months. Furthermore, a small number of samples came from children with a positive HIV status at 18 months (n=4). As the aim of the specific HIV analyses was to compare between CHEU vs CHU, children require a known HIV status in order to be classified as children HIV-exposed uninfected (CHEU) or children HIV-unexposed (CHU). This is outlined in the methods (Page 34-35). The authors have added a sentence in the results (Page 10, Line 10-12) to clarify the sample size for the CHEU vs CHU analysis (n=847).

The acronyms CHU/CHEU are very similar to each other and are unintuitive. For example, CHU (children HIV exposed) should instead be HUC (HIV exposed children). A much more readable alternative that would differentiate well the two groups is HIVExp- and HIVExp+ Inf-

The authors acknowledge the confusion added by the HIV exposure acronyms. Following advice from co-authors in the HIV research field, the authors have used the current and most appropriate 'person-first' terminology for children exposed or unexposed to HIV. This terminology is preferred and advised in the HIV community in order to destigmatize those living with or exposed to HIV. The authors feel that introducing another term could result in additional confusion in this evolving field.

While it is mentioned in the Discussion that cotrimoxazole treatment might impact the results >6 weeks, the authors should expand on the potential impact of this broad-spectrum antibiotic on the gut microbiome assembly and maturation.

The authors have amended the discussion (Page 23) to expand on the potential influence of cotrimoxazole in the context of previous research.

Figure 5 and 6 have identical caption title.

Thank you for highlighting this. The authors have now corrected this figure title.

Figure 5 and 6: specify that "HIV-positive" and "HIV-negative" refer to the maternal status.

Both figures have been amended for clarification.

Page 3, Line 7: "In addition to research investigating"

The authors are unsure to what this comment refers.

LAZ and WHZ acronyms are described in the Methods (page 26) but are mentioned as early as page 6. Add at the first reference a pointer to the detailed description and specify that the acronyms are scores.

The LAZ and WHZ acronyms have now been outlined on Page 6 and Page 12.

Page 15, Line 20 and Page 16, line 14: "through the first 18 months after birth" implies that sampling was performed during the whole duration of the 18 months (starting from birth), which is not the case.

These sentences have been amended to state "*...from 1-18 months of age*"

Reviewer comments, second round review –

Reviewer #1 (Remarks to the Author):

The revised manuscript is much improved, and all my concerns have been properly addressed. I'd like to recommend it to be accepted after the authors make one minor change.

1. The font size of figures 1, 3, 5 and 6 is too small, and it should be enlarged.

Reviewer #2 (Remarks to the Author):

I am satisfied that all issues raised by me have been addressed in the revised manuscript.

Reviewer #4 (Remarks to the Author):

Results:

Last paragraph in page 7 starts talks about XGBoost for Age prediction. It is understood that chronologic age is used as the outcome. It appears that authors built a model on the data from normal children (HIV- mothers, LAZ>-2), training data. The model was then tested in the remaining. The remaining hypothetically would include A) Stunted and mother HIV+, B) Stunted and mother HIV-, C) not stunted and mother HIV+. First of all, it would be super helpful for readers to see n of child and n of sample in both training and test sets, in the same paragraph. Second, the model learns from normal children. Microbiome (MB) wise, it is ok however, the model never learns how to account for mother HIV status to predict age. This is not a major critique, however, why not to build a model to predict age on all non-stunted children, and add mother HIV status as a predictor, and test it on stunted children?

Methods:

in stage 1, 10-fold CV was used during Bayesian Hyperparameter optimization. In stage 2, when epidemiological variables were added, leave-one-out was used. Why? Why two different CV approach in two stages?

"This 3-stage hyperparameter tuning and model 18 building was performed for two feature sets, one comprising microbiome features and a 19 second comprising microbiome plus epidemiologic features;": Not clear if the microbiome (MB) features identified in Stage 1 or Stage 2. It reads as Stage 2. If this is the case (please just clarify in the text if Stage 1), it may not be a good approach to compare MB vs MP+EPI variables. It is because, in stage 2, the model accounted for both MB of Stage 1 and EPI variables. Some MB variables may show lower importance just because some EPI variables explains the variability that could be explained by some MBs.

Microbiome Age.

* "Leave-one-out cross validation was used to build a model on the full dataset (n = 859), and model performance was computed using the observed and predicted model response for the "healthy training set". Model performance for the "healthy" and "unhealthy" groups was then computed using the observed responses and the cross-validation hold-out predictions.". First, entire dataset was used to build a model. Doing it with leave-one-out doesn't avoid overfitting. It is very much likely that the model is overfitted and may not yield to any clinically meaningful conclusion. What hold out is being referred to? The 'one' sample left out during leave-one-out process? A better setup could be to split data into model building and hold out, do the CV on the model building data, and then test on the hold-out.

Fig S4. Why MAE decreases as age increases? If it was a traditional regression, we would suspect heteroskedasticity. Yet, the results are coming from XGBoost. These visuals in this figure needs

additional discussion.

Overall, the cross validation process needs clarifications and/or improvements.

REVIEWER COMMENTS

Reviewer #1 (Remarks to the Author):

The revised manuscript is much improved, and all my concerns have been properly addressed. I'd like to recommend it to be accepted after the authors make one minor change.

1. The font size of figures 1, 3, 5 and 6 is too small, and it should be enlarged.

The authors would like to thank Reviewer 1 for their comments and input which have helped to improve our manuscript. We have increased the font size of the relevant figures as much as possible in order to improve legibility.

Reviewer #2 (Remarks to the Author):

I am satisfied that all issues raised by me have been addressed in the revised manuscript.

The authors would like to thank Reviewer 2 for their comments and input which have helped to improve our manuscript.

Reviewer #4 (Remarks to the Author):

1. Results:

Last paragraph in page 7 starts talks about XGBoost for Age prediction. It is understood that chronologic age is used as the outcome. It appears that authors built a model on the data from normal children (HIV- mothers, LAZ>-2), training data. The model was then tested in the remaining. The remaining hypothetically would include A) Stunted and mother HIV+, B) Stunted and mother HIV-, C) not stunted and mother HIV+. First of all, it would be super helpful for readers to see n of child and n of sample in both training and test sets, in the same paragraph. Second, the model learns from normal children. Microbiome (MB) wise, it is ok however, the model never learns how to account for mother HIV status to predict age. This is not a major critique, however, why not to build a model to predict age on all non-stunted children, and add mother HIV status as a predictor, and test it on stunted children?

We agree with the reviewer these analyses could be performed in two different ways. Either using maternal HIV status as a predictor, or applying the prediction model to the stunted groups separated by maternal HIV status. HIV exposure in utero was strongly associated with stunting and other outcomes in the SHINE trial and all analyses as part of the SHINE trial were stratified by maternal HIV status (1-4). We chose non-stunted, non-HIV-exposed children as a cohort of children representing optimal 'health' in order to

model ‘healthy’ microbiome maturation in this context. We also chose to limit the model to microbiome features for this microbiome age analysis. Although adding maternal HIV status as a predictor would allow the model to account for HIV status when examining the relationship between microbiome age and stunting, other factors would also have to be included in this prediction model (including maternal height, birthweight, breast-feeding status etc). We have revised the results section to include the number of samples and number of infants in each of the training and test sets in the specified paragraph.

2. Methods:

In stage 1, 10-fold CV was used during Bayesian Hyperparameter optimization. In stage 2, when epidemiological variables were added, leave-one-out was used. Why? Why two different CV approach in two stages?

We agree with the reviewer that the text is confusing. Epidemiologic variables were not added in stage 2, but rather two models were run starting at stage one (a model containing microbiome features only and a second including microbiome + epidemiologic features). We have revised the methods section to clarify our 3-stage model building approach (manuscript page 34).

“XGBoost model selection was performed in 3 stages as previously described (30). In brief, BayesianOptimization function of the rBayesianOptimization package was used with 10-fold cross-validation to select model hyperparameters by minimizing the mean squared error (MSE). This 3-stage hyperparameter tuning and model building was performed for two feature sets, one comprising microbiome features and a second comprising microbiome plus epidemiologic features; this was done to assess model performance and to examine the contribution of epidemiologic versus microbiome features. In stage one, the model dataset included either microbiome features alone or microbiome and epidemiologic features. Models with the lowest MSE (in the 5th percentile) were retained, and from these models the variables that contributed to the top 95% of variable importance by proportion were retained; this filtering was applied separately to the microbiome and epidemiologic features. In stage two, models were built with the retained features obtained from stage one, and BayesianOptimization was re-run as for stage one but using leave-one-out cross-validation. Models containing microbiome features alone, or together with epidemiologic features, were again filtered according to feature importance as for stage one. In stage three, all retained epidemiologic variables, microbiome features, and hyperparameters selected in stage two were used to fit our final models, using leave-one-out cross-validation to minimize the MSE.”

We did implement 10-fold CV in stage one, due to the number of features in the datasets and the number of infant growth models (varying by feature set, age group, maternal HIV status and outcome type). LOOCV is time intensive, especially given our analysis approach. Moreover, Bayesian Optimization requires multiple iterations of model building. In our case, we started with an initial set of 10 points in the hyperparameter space and built a model for each one; this was followed by 50 iterations of the Bayesian method, where each iteration requires the building of another model. Given the computational requirements of this approach, we chose to use 10-fold CV in stage one.

3. “This 3-stage hyperparameter tuning and model 18 building was performed for two feature sets, one comprising microbiome features and a 19 second comprising microbiome plus epidemiologic features;”: Not clear if the microbiome (MB) features identified in Stage 1 or Stage 2. It reads as Stage 2. If this is the case (please just clarify in the text if Stage 1), it may not be a good approach to compare MB vs MP+EPI variables. It is because, in stage 2, the model accounted for both MB of Stage 1 and EPI variables. Some MB variables may show lower importance just because some EPI variables explains the variability that could be explained by some MBs.

We have revised the methods text to clarify model building and feature selection, please see response above.

4. Microbiome Age.

* “Leave-one-out cross validation was used to build a model on the full dataset (n = 859), and model performance was computed using the observed and predicted model response for the “healthy training set”. Model performance for the “healthy” and “unhealthy” groups was then computed using the observed responses and the cross-validation hold-out predictions.”. First, entire dataset was used to build a model. Doing it with leave-one-out doesn’t avoid overfitting. It is very much likely that the model is overfitted and may not yield to any clinically meaningful conclusion.

There was a manuscript editing error. The LOOCV was used to build a model on the ‘healthy training set’, not on the full dataset. We have revised the methods section to correct and clarify the text (manuscript pages 35-36). We have substituted ‘out-of-fold’ for ‘hold-out’, as this is a more commonly used term. We aggregated the out-of-fold predictions to yield a vector of predicted values, and used this vector, along with observed values, to compute the model performance.

“We performed the same 3-stage tuning and model building procedure, as described above. We generated model performance metrics, including pseudo-R2, mean absolute error (MAE) and mean squared error (MSE) for the three sets. Leave-one-out cross validation was used to build a model on the ‘healthy training set’ (n = 265 samples). Model performance for the ‘healthy training set’ was computed using the observed response and cross validation out-of-fold predictions. Model performance for the ‘healthy’ and ‘unhealthy’ test sets was computed using the observed responses and the final model predictions.”

5. What hold out is being referred to? The ‘one’ sample left out during leave-one-out process? A better setup could be to split data into model building and hold out, do the CV on the model building data, and then test on the hold-out.”

An attractive alternative to train/test splits for model validation is model prediction using out-of-fold observations, especially in the context of small sample size, which was the case for our models. For each observation a model is built using all remaining observations. The initial one observation is referred to as the out-of-fold observation. The model is then used to predict the response for that one out-of-fold observation. The collection of

predicted responses is then used to compute the model performance metrics. We were concerned that splitting the 'healthy training set' into training and testing sets would increase model error, due to smaller sample sizes in each split. Therefore, we relied on the out-of-fold predictions to evaluate model performance, which we believed to be a more suitable approach.

6. Fig S4. Why MAE decreases as age increases? If it was a traditional regression, we would suspect heteroskedasticity. Yet, the results are coming from XGBoost. These visuals in this figure needs additional discussion.

The reviewer is correct to note that MAE decreases as age increases, and as noted, these results come from XGBoost prediction models rather than traditional regression. Predictions of growth were run as separate models for each age groups. Therefore, we would not assume heteroskedasticity fully explains this trend in decreasing MAE with age. Reasons for the decreasing MAE are unclear; however, we believe that the gut microbiome of 12-18 month old children is better at predicting growth than at 1-6 months of age, thereby producing less error in our XGBoost models. Our own data (Figure S3 c-d) and previous data (5) show far less inter-individual variability in 12-18 month gut microbiomes versus 1-6 months. This reduced inter-individual variability suggests that associations with growth are more sensitive to variations in microbiome composition/function between 12-18 months of age. We have amended the discussion to discuss this as suggested by the reviewer.

7. Overall, the cross validation process needs clarifications and/or improvements.

We are grateful to the reviewer for prompting us to clarify our cross-validation methods. As mentioned above, we have corrected and clarified the cross-validation text in the manuscript. As suggested by the editor, we have also added a discussion at the beginning of the limitations section describing the potential risks of over-fitting using such models, which could thereby influence the interpretability of our results.

References

1. Humphrey JH, Mbuya MNN, Ntozini R, Moulton LH, Stoltzfus RJ, Tavengwa NV, et al. Independent and combined effects of improved water, sanitation, and hygiene, and improved complementary feeding, on child stunting and anaemia in rural Zimbabwe: a cluster-randomised trial. *Lancet Glob Health*. 2019;7(1):e132-e47.
2. Prendergast AJ, Chasekwa B, Evans C, Mutasa K, Mbuya MNN, Stoltzfus RJ, et al. Independent and combined effects of improved water, sanitation, and hygiene, and improved complementary feeding, on stunting and anaemia among HIV-exposed children in rural Zimbabwe: a cluster-randomised controlled trial. *The Lancet Child & Adolescent Health*. 2019;3(2):77-90.
3. Gladstone MJ, Chandna J, Kandawasvika G, Ntozini R, Majo FD, Tavengwa NV, et al. Independent and combined effects of improved water, sanitation, and hygiene (WASH) and improved complementary feeding on early neurodevelopment among children born to HIV-negative mothers in rural Zimbabwe: Substudy of a cluster-randomized trial. *PLoS Med*. 2019;16(3):e1002766.

4. Ntozini R, Chandna J, Evans C, Chasekwa B, Majo FD, Kandawasvika G, et al. Early child development in children who are HIV-exposed uninfected compared to children who are HIV-unexposed: observational sub-study of a cluster-randomized trial in rural Zimbabwe. *J Int AIDS Soc.* 2020;23(5):e25456.
5. Bäckhed F, Roswall J, Peng Y, Feng Q, Jia H, Kovatcheva-Datchary P, et al. Dynamics and Stabilization of the Human Gut Microbiome during the First Year of Life. *Cell Host Microbe.* 2015;17(5):690-703.

Reviewer comments, third round review –

Reviewer #4 (Remarks to the Author):

Thank you for the edits and clarifications